# UV-B irradiation-activated E3 ligase GmILPA1 modulates gibberellin catabolism to increase plant height in soybean

Jiaqi Sun[1], Shiyu Huang[1], Qing Lu[1], Shuo Li[1], Shizhen Zhao[1], Xiaojian Zheng[1], Qian Zhou[1], Wenxiao Zhang[1], Jie Li[1], Lili Wang[1], Ke Zhang[1], Wenyu Zheng[1], Xianzhong Feng[2] ✉, Baohui Liu[3] ✉, Fanjiang Kong[3] ✉ & Fengning Xiang[1] ✉

Plant height is a key agronomic trait that affects yield and is controlled by both phytohormone gibberellin (GA) and ultraviolet-B (UV-B) irradiation. However, whether and how plant height is modulated by UV-B-mediated changes in GA metabolism are not well understood. It has not been reported that the E3 ubiquitin ligase Anaphase Promoting Complex/Cyclosome (APC/C) is involved in the regulation of plant growth in response to environmental factors. We perform a forward genetic screen in soybean and find that a mutation in *Glycine max Increased Leaf Petiole Angle1* (*GmILPA1*), encoding a subunit of the APC/C, lead to dwarfism under UV-B irradiation. UV-B promotes the accumulation of GmILPA1, which ubiquitinate the GA catabolic enzyme GA2 OXIDASE-like (GmGA2ox-like), resulting in its degradation in a UV-B-dependent manner. Another E3 ligase, GmUBL1, also ubiquitinate GmGA2ox-like and enhance the GmILPA1-mediated degradation of GmGA2ox-like, which suggest that GmILPA1-GmGA2ox-like module counteract the UV-B-mediated reduction of bioactive GAs. We also determine that *GmILPA1* is a target of selection during soybean domestication and breeding. The deletion (Indel-665) in the promoter might facilitate the adaptation of soybean to high UV-B irradiation. This study indicates that an evolutionary *GmILPA1* variant has the capability to develop ideal plant architecture with soybean cultivars.

Ideotype breeding aims to achieve high yields in crops by combining multiple beneficial genetic traits into one variety. Plant height is an important feature of plant ideotypes. The application of semi dwarf varieties has significantly improved crop yield by increasing the planting density and the lodging resistance[1,2]. The gains in grain productivity during the Green Revolution were a direct consequence of optimal plant height[3]. Mutant alleles of the Green Revolution genes

*Semidwarf1* (*Sd1*) and *Reduced height* (*Rht*) are utilized to improve crop yields by decreasing overall plant height and increasing the harvest index[4]. Although dozens of quantitative trait loci (QTLs) have been identified that control plant height in soybean (*Glycine max* L.), few of the causal genes have been isolated and characterized[5,6]. For example, *GmDW1* (*DWARF 1*) encodes an ent-kaurene synthase[7], *GmLHY* (*LATE ELONGATED HYPOCOTYL*) encodes an MYB transcription factor,

[1]The Key Laboratory of Plant Development and Environmental Adaptation Biology, Ministry of Education, School of Life Sciences, Shandong University, Qingdao 266237, China. [2]Key Laboratory of Soybean Molecular Design Breeding, Northeast Institute of Geography and Agroecology, Chinese Academy of Sciences, Changchun 130000, China. [3]Guangdong Key Laboratory of Plant Adaptation and Molecular Design, Guangzhou Key Laboratory of Crop Gene Editing, Innovative Center of Molecular Genetics and Evolution, School of Life Sciences, Guangzhou University, Guangzhou 510006, China. ✉e-mail: fengxianzhong@iga.ac.cn; liubh@gzhu.edu.cn; kongfj@gzhu.edu.cn; xfn0990@sdu.edu.cn

*Determinate stem 1* (*Dt1*) is a homolog of *TERMINAL FLOWER 1*, and *Dt2* encodes dominant MADS domain factor[8–10] could help deploy the semi-dwarf phenotype and improve soybean yields.

The plant hormone gibberellins play a pivotal role in determining plant height[11,12]. Mutations in genes involved in GA biosynthesis, metabolism, and signaling influence plant height. Key components of the GA signaling pathway have been identified from genetic screens performed in rice (*Oryza sativa*) and *Arabidopsis* (*Arabidopsis thaliana*). For instance, DELLA proteins restrain plant growth, and their degradation upon perception of GA signal promotes growth[13]. Loss-of-function mutations in GA biosynthetic genes such as *GA20ox* and *GA3ox* lead to dwarfism[14–19]. Conversely, GA2ox members are GA catabolic enzymes that inactivate the phytohormone and negatively regulate plant growth, and the overexpression of their encoding genes suppresses stem growth[20–24]. Several transcription factors can also negatively regulate GA-mediated stem cell elongation by activating *GA2ox* transcription[25,26]. Furthermore, *GA2ox* expression is induced by environmental signals, including salt stress, high temperature, and light[27–30]. While much is known about the transcriptional modulation of *GA2ox*s, whether and how GA2oxs are post-translationally regulated are largely unknown.

Exposure to ultraviolet-B (UV-B) light is an abiotic stressor that affects plant height[31,32]. In *Arabidopsis*, inhibition of hypocotyl growth involves the transcriptional activation of *ELONGATED HYPOCOTYL5* (*HY5*) and *HY5 HOMOLOG* (*HYH*) by the UV-B photoreceptor UV RESISTANCE LOCUS8 (UVR8), HY5/HYH stability, and the degradation of the transcription factors PHYTOCHROME-INTERACTING FACTOR4 (PIF4) and PIF5[32–35]. In addition, UV-B also inhibits plant growth in *Arabidopsis* by promoting the accumulation of DELLA proteins, thus repressing GA signaling output[36]. UV-B exposure also decreases the transcript levels of GA biosynthesis genes, such as *GA3ox* and *GA20ox*, and increases the transcript levels of *GA2ox*[37] to inhibit hypocotyl growth. To date, UV-B has only been shown to affect the transcription of GA biosynthesis and catabolism genes.

The ubiquitin-proteasome system (UPS) is a major regulatory mechanism for the selective degradation of proteins in eukaryotes[38] and plays a key role in plant growth and development[39–43]. Ubiquitin is attached to substrate proteins by a series of enzymatic reactions. First, ubiquitin molecules are activated by an E1 activating enzyme in an ATP-dependent manner, followed by its transfer to an E2 conjugating enzyme, and finally covalent attachment to the target protein by an E3 ubiquitin ligase[44]. Ubiquitin tags label target proteins for degradation via the 26S proteasome using its ATP-dependent endopeptidase activity[45]. Each E3 ubiquitin ligase strictly controls the efficiency and substrate specificity of the UPS. E3 ubiquitin ligases are divided into four types: HECT (Homology to E6AP C Terminus), RING/U-box (Really Interesting New Gene/U-Box), SCF (Skp1, Cullin, and F-box protein complex), and APC/C (anaphase promoting complex/cyclosome)[46]. Of these, the APC/C is a conserved multi-subunit ubiquitin ligase assembled from 13 different subunits and plays an important role during the cell cycle[47]. The APC/C is also involved in plant growth and development, modulating plant height, leaf size, lateral root number, and silique development[48]. In addition, APC/C activity was reported to influence phytohormone signals, as auxin distribution is disrupted in embryos of *apc4* mutants[49]. In rice, the *apc6* mutant has a shorter stature and does not respond to exogenous GA[50]. However, APC/C-mediated regulation of plant growth and development by environmental factors has not been reported.

In this study, we identified a soybean dwarf mutant in a forward genetic screen. The *Gmuid1* (soybean *UV-B induced dwarfism1*) mutant showed a dwarf phenotype in a UV-B-dependent manner. And we found that the mutation in *Glycine max Increased Leaf Petiole Angle1* (*GmILPA1*), encoding an APC8-like protein, resulted in the dwarfism of *Gmuid1* under UV-B irradiation. We further demonstrated that UV-B induces the accumulation of GmILPA1, which reduced UV-B-induced

growth inhibition by interacting with GmGA2ox-like and targeting it for degradation in response to UV-B. Another E3 ubiquitin ligase, GmUBL1, interacted with both GmILPA1 and GmGA2ox-like, respectively, to promote degradation of GmGA2ox-like. Our findings indicated that GmILPA1 promotes GA catabolism to modulate plant height under UV-B exposure. Our study characterized the effect of UV-B exposure and GA application on plant height, with a view to gain a better understanding of the mechanism of APC/C by UV-B-mediated dwarfism in soybean. Importantly, *GmILPA1* underwent artificial selection during soybean domestication, with the four major haplotypes *GmILPA1*[Hap1], *GmILPA1*[Hap2], *GmILPA1*[Hap5] and *GmILPA1*[Hap6] having been targeted by selection in improved cultivars. The shorter height of plants harboring *GmILPA1*[Hap5] might be due to the specific mutations in cis-regulatory elements affecting the response to UV-B.

## Results

### GmILPA1 is a positive regulator of plant height in soybean

To isolate novel genes that regulate plant height in soybean, we performed a forward genetic screen of the Chinese soybean cultivar Hedou 12 (H12) mutagenized with ethyl-methanesulfonate (EMS). We identified the mutant *Gmuid1* in M4 populations based on its shorter plant height and internode length when grown in the field compared to H12 plants (Fig. 1a–c). *Gmuid1* plants also had smaller leaves, shorter petioles, and a larger leaf petiole angle than in H12 (Supplementary Fig. 1a, b). We investigated the underlying cellular basis of the short internodes of *Gmuid1* by measuring cell length in *Gmuid1* and H12 plants. Compared to H12, longitudinal parenchyma cells in *Gmuid1* internodes were significantly shorter (Supplementary Fig. 1c, d). Furthermore, we observed no significant difference in the number of nodes along the main stem and branches between H12 and *Gmuid1*; however, the other agronomic traits, including number of pods, number of seeds per plant, grain weight per plant, and hundred-grain weight, were all lower in *Gmuid1* than in H12 (Supplementary Fig. 1e–j).

To identify the causative gene responsible for the observed phenotypic changes, we crossed the soybean cultivar Williams 82 to the *Gmuid1* mutant, yielding an F2 population comprising 201 individual F2 plants. Of these F2 plants, 153 showed the wild-type phenotype and 48 exhibited the mutant phenotype, thus fitting a 3:1 segregation ratio ($\chi^2$ test, $P = 0.89$). This result indicated that the *Gmuid1* mutation affects a single gene and is recessive. We used 128 INDEL markers and the 48 individual F2 plants to map the *GmUID1* locus, the results indicated that the mutant locus was mapped to a 4.5-Mb genomic region between markers Gm1005 and Gm0069 on chromosome 11, based on the soybean reference genome from Williams 82[51]. We then developed new molecular markers within this interval and genotyped a set of 280 individual F2 plants with the mutant phenotype from new F1 inbred offspring, which allowed us to narrow down the interval to a 0.72-Mb region between INDEL markers Gm1008 and Gm1009. We further developed F3 lines from F2 lines homozygous for the *Gmuid1* mutation but heterozygous at one of the flanking markers, culminating in a 68-kb region between SSR markers Gm1015 and Gm1016 based on 2250 F2 and F3 individuals. This interval contains six open reading frames in the Williams 82 reference genome[51] (Fig. 1d). Independently, we performed a bulk segregant analysis using two sets of pooled F2 plants exhibiting the mutant or wild-type (WT) phenotype (a mix of homozygous and heterozygous plants for the WT allele) to identify all single nucleotide polymorphisms (SNPs) relative to Williams 82 and calculate the SNP index for each pool. The sequencing depth of each pool was 40× genome coverage. This analysis highlighted a region near the top of chromosome 11 (Supplementary Fig. 2a-b), which was consistent with the map-based cloning above. Importantly, we identified one SNP (G-to-A at position 222 of genomic DNA) in the splicing site of the first intron for candidate gene *Glyma.11G026400.1* in the *Gmuid1* mutant; the remaining five candidate genes showed no SNPs between H12 and Williams 82 (Fig. 1d). The single nucleotide substitution at the splicing

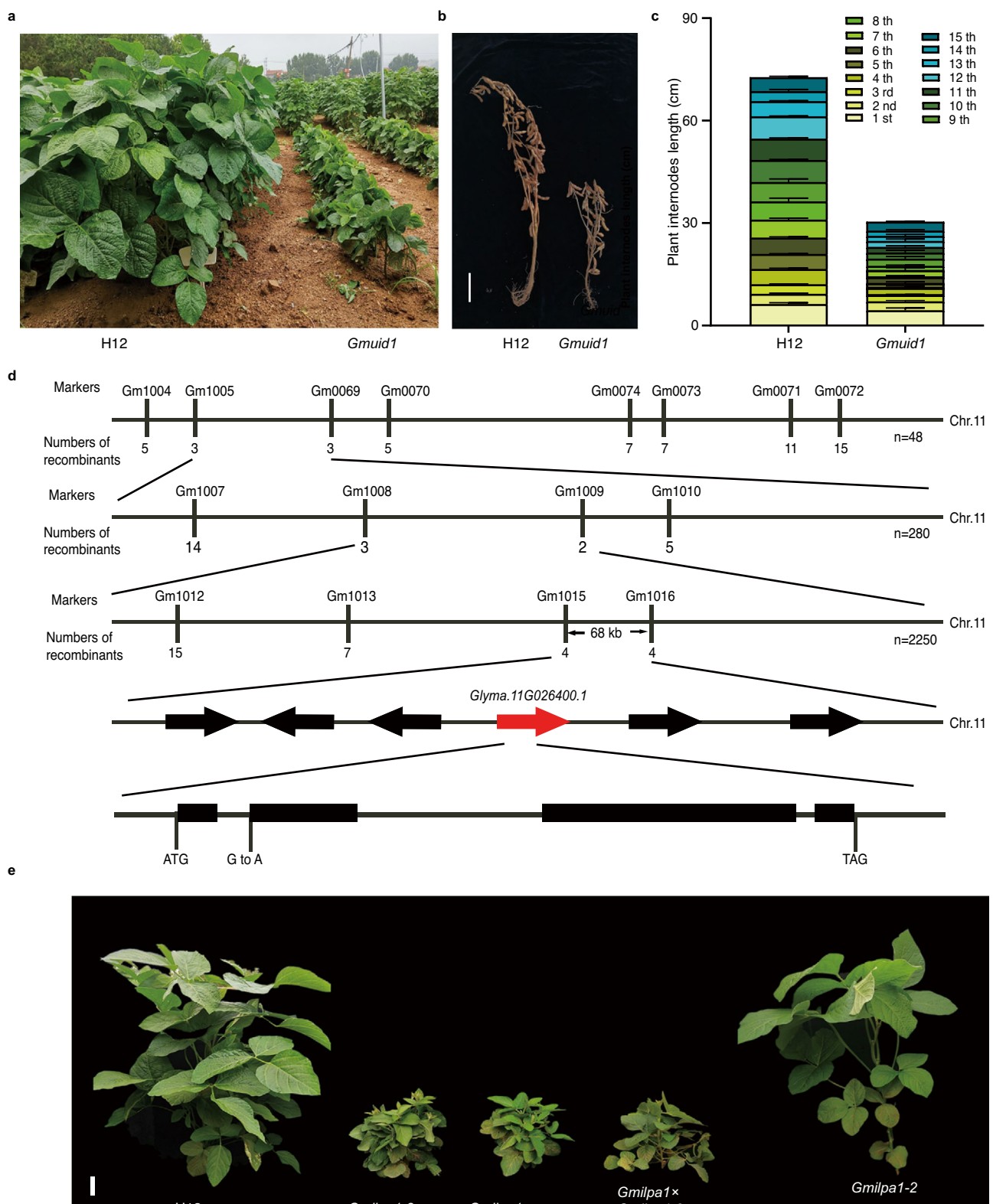

**Fig. 1 | Identification and characterization of the dwarf mutant *Gmuid1*.**
**a** Representative image of plants from the wild-type (WT) Hedou 12 (H12) and the *Gmuid1* mutant at the vegetative stage in the field. **b** Representative images of the WT and *Gmuid1* at the R8 stage (full maturity). Scale bar, 10 cm. **c** Internode length of WT and *Gmuid1* plants at the R8 stage. Data are means ± standard deviation (SD), $n = 6$ independent plants. **d** Summary of the positional cloning of the *GmUID1* (also named *GmILPA1*) locus to chromosome 11. The insertion/deletion (InDel) markers Gm1005 and Gm0069 were used for initial mapping. *GmUID1* (*GmILPA1*) was fine-mapped to a 68-kb region between the single sequence repeat (SSR) markers Gm1015 and Gm1016 on chromosome 11. *n* represents individual $F_2/F_3$ plants with the mutant phenotype. **e** Representative images of H12, *Gmilpa1-2* (*Gmuid1*), *Gmilpa1*, *Gmilpa1-2 × Gmilpa1* F1, and *Gmilpa1-2 35S:GmILPA1* plants at the R1 stage (beginning of flowering). Scale bar, 10 cm. Source data are provided as a Source Data file.

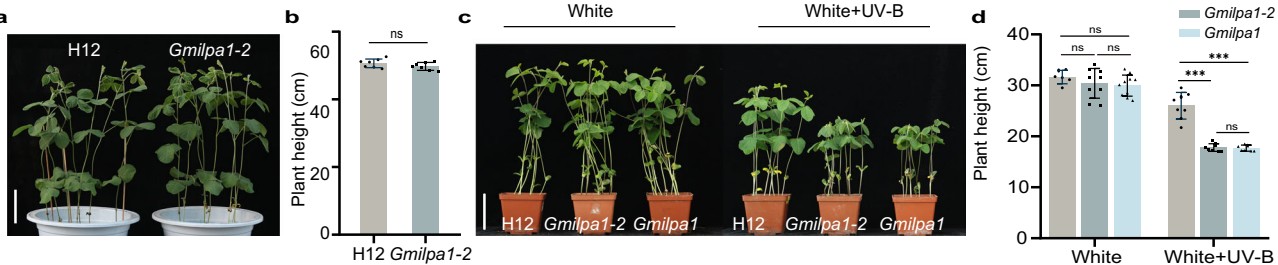

**Fig. 2 | UV-B is required for the *Gmilpa1-2* phenotype. a** Representative image of the WT Hedou 12 (H12) and *Gmilpa1-2* grown in a glasshouse at the V4 stage (four unrolled trifoliate leaves). Scale bar, 10 cm. **b** Height of the plants shown in (**a**), ($n = 7$ biologically independent samples; $P = 0.1909$), $P$-values were calculated by comparing to H12 plants using two-tailed Student's $t$-test; ns not significant. **c** Representative images of the V2 stage of H12, *Gmilpa1-2*, and *Gmilpa1* grown under white light (600 μmol m$^{-2}$ s$^{-1}$ white light, no UV-B, 25 °C, and a 12-h-light/12-h-dark photoperiod) and white light supplemented with UV-B (600 μmol m$^{-2}$ s$^{-1}$ white light, 1.5 μmol m$^{-2}$ s$^{-1}$ UV-B, 25 °C, and a 12-h-light/12-h-dark photoperiod). Scale bar, 10 cm. **d** Height of the seedlings shown in (**c**), ($n > 6$ biologically independent samples; $P = 2.42 \times 10^{-6}$ in H12 and *Gmilpa1-2*, $P = 1.80 \times 10^{-6}$ in H12 and *Gmilpa1*). Data are presented as mean values ± SD; $P$-value is calculated with a one-way ANOVA analysis−Tukey comparison, and the columns labeled without the same alphabet are significantly different ($P < 0.05$, two-sided). ***$P < 0.001$, ns not significant. Source data are provided as a Source Data file.

site altered messenger RNA (mRNA) splicing and led to premature termination in the new increased exon of the transcript encoded by *Glyma.11G026400* in the *Gmuid1* mutant (Supplementary Fig. 3). Therefore, we identified a candidate gene for regulating plant height.

Additionally, *Glyma.11G026400* was previously reported and named *GmILPA1* (*Glycine max Increased Leaf Petiole Angle1*)[52]; the mutant *Gmilpa1* resulted in the deletion of exon 4 of *Glyma.11G026400*[52,53] (Supplementary Fig. 4a). Similar to *Gmuid1*, the *Gmilpa1* mutant also showed a dwarf phenotype in the field (Fig. 1e, Supplementary Fig. 4b). As expected, F$_1$ plants from the cross between *Gmuid1* and *Gmilpa1* also exhibited the dwarf phenotype (Fig. 1e, Supplementary Fig. 4c, d). Therefore, *Gmuid1* is an allelic mutant of *Gmilpa1*, we renamed the *Gmuid1* as *Gmilpa1-2*. In addition, we obtained transgenic lines overexpressing its coding sequence cloned in-frame with that of the green fluorescent protein gene (*GFP*) in the *Gmilpa1-2* mutant background. We obtained six independent transformation events; their T$_2$ progeny accumulated GFP and restored the mutant phenotype to the WT, as evidenced by plant height (Fig. 1e and Supplementary Fig. 4e–f). Collectively, all these results indicated that *Glyma.11G026400* is *GmUID1* (*GmILPA1*), which controls plant height in soybean.

Sequence comparison and phylogenetic analysis demonstrated that *GmILPA1* encodes an APC8-like protein, a subunit of the APC/C (Supplementary Fig. 5a). Reverse transcription quantitative PCR (RT-qPCR) analysis indicated that *GmILPA1* is constitutively expressed in roots, hypocotyls, cotyledons, stems, leaves, flowers, and pods (Supplementary Fig. 5b). We also transiently expressed a *GmILPA1-GFP* fusion construct driven by the cauliflower mosaic virus (CaMV) 35S promoter in *Nicotiana benthamiana* leaves, in agreement with previous report[53], we also found that GmILPA1-GFP localizes to both the nucleus and cytoplasm (Supplementary Fig. 5c).

### *GmILPA1* promotes plant height in a UV-B-dependent manner

To further characterize the *Gmilpa1-2* mutant, we performed phenotypic analyses in a glasshouse. To our surprise, *Gmilpa1-2* plants did not have a dwarf phenotype under these conditions (Fig. 2a, b). Considering the differences between the light spectra of sunlight and white light-emitting diodes (LEDs)[54], we hypothesized that the differential phenotype of *Gmilpa1-2* might be caused by the lack of UV-B light from LEDs. To test this hypothesis, we grew H12, *Gmilpa1-2* and *Gmilpa1* seedlings under white light (600 μmol m$^{-2}$ s$^{-1}$ white light, no UV-B, 25 °C, and a 12-h-light/12-h-dark photoperiod) and white light supplemented with UV-B (600 μmol m$^{-2}$ s$^{-1}$ white light, 1.5 μmol m$^{-2}$ s$^{-1}$ UV-B, 25 °C, and a 12-h-light/12-h-dark photoperiod) to the V2 stage. We observed shorter plants with *Gmilpa1-2* and *Gmilpa1* relative to H12 in

response to UV-B irradiation (Fig. 2c, d). We concluded that UV-B is essential for evoking the dwarf phenotype of the *Gmilpa1* mutant.

### UV-B promotes the interaction between GmILPA1 and GmGA2ox-like

To investigate how GmILPA1 regulates plant height in soybean, we performed a yeast two-hybrid (Y2H) screen with a soybean cDNA library to identify GmILPA1-interacting proteins. Among the potential candidate interactors, we noticed two proteins with potential roles in modulating plant height. Indeed, one positive clone encodes 2-oxoglutarate−dependent dioxygenase (2ODD), which is a close homolog of *Arabidopsis* GA2ox4 that we named GmGA2ox-like (Supplementary Fig. 6a). To confirm GmGA2ox-like is a functional GA2-oxidase, we performed the in vitro enzymatic activity assay using GA$_1$ and GA$_4$ as the substrates. The results showed that GmGA2ox-like converted GA$_1$ and GA$_4$ to their corresponding 2β-hydroxylated products GA$_8$ and GA$_{34}$, respectively (Supplementary Fig. 6b, c). Next, we generated 3 lines overexpressing *GmGA2ox-like*, which displayed a dwarf phenotype compared to non-transgenic WT plants (Supplementary Fig. 6d, e). Sequence analysis revealed that GmGA2ox-like harbors a typical D-box (Supplementary Fig. 6f). The second clone encodes a protein with a RING/U-box domain that is predicted to have E3 ubiquitin ligase activity; we named this protein GmUBL1 (ubiquitin ligase 1). To validate these interactions, we carried out yeast two-hybrid (Y2H) assays. The Y2H assay showed that GmILPA1 interacts with GmGA2ox-like and GmUBL1, respectively (Fig. 3a). To confirm these interactions in vivo, we performed bimolecular fluorescence complementation (BiFC) and co-immunoprecipitation (Co-IP) assays in *N. benthamiana* leaves transiently expressing the appropriate pairs of constructs. We detected yellow fluorescence protein (YFP) fluorescence signals in *N. benthamiana* epidermal cells co-expressing *GmILPA1-YFP$^C$* (encoding a fusion between GmILPA1 and the C-terminal half of YFP) and *GmGA2ox-like-YFP$^N$* (encoding a fusion between GmGA2ox-like and the N-terminal half of YFP) or *GmILPA1-YFP$^C$* and *GmUBL1-YFP$^N$* (Fig. 3b). We also confirmed these protein interactions by Co-IP assays as well (Fig. 3c, d, Supplementary Fig. 7c). We asked whether GmGA2ox-like and GmUBL1 might also interact. We explored this possibility with Y2H, BiFC, and Co-IP assays. Indeed, we determined that GmGA2ox-like also interacts with GmUBL1 with all three assays (Fig. 3a, b, e). Together, our results demonstrated that GmILPA1, GmGA2ox-like, and GmUBL1 interact with each other. In addition, GmILPA1 and GmGA2ox-like, and GmILPA1 and GmUBL1 were co-localized, respectively, in the cytoplasm and nucleus by transient expression in *N. benthamiana* leaves (Supplementary Fig. 8a, b).

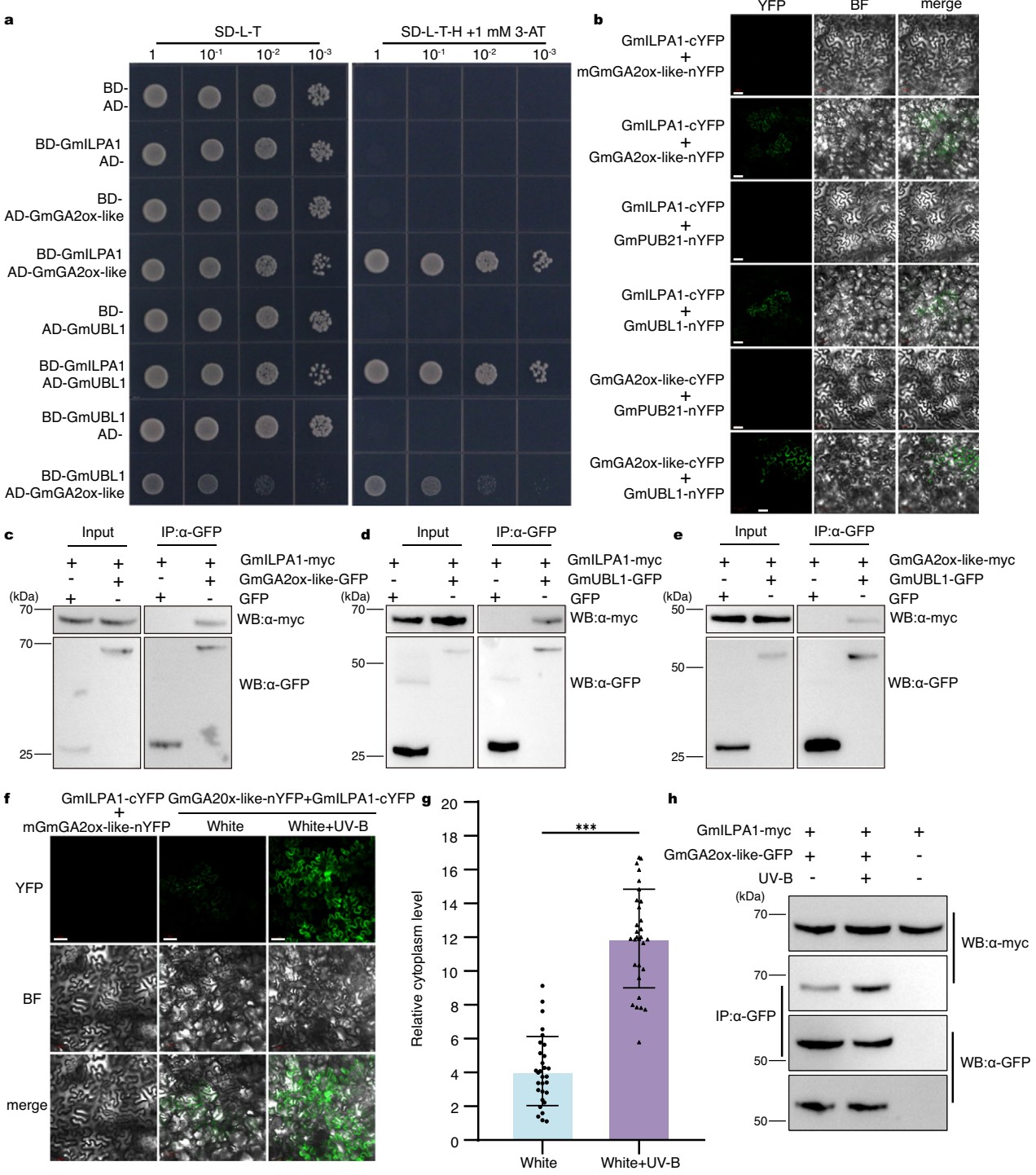

As *Gmilpa1-2* showed a dwarf phenotype only upon UV-B exposure, we investigated the potential effect of UV-B on the interaction between GmILPA1, GmGA2ox-like, and GmUBL1. BiFC assays indicated that UV-B treatment promotes the interaction between GmILPA1 and GmGA2ox-like in the cytoplasm, as evidenced by the three-fold higher relative fluorescence intensity detected in the cytoplasm following treatment with UV-B relative to controls maintained under white light throughout (Fig. 3f, g). We further tested the effect of UV-B on the interaction of GmILPA1 and GmGA2ox-like using Co-IP assays. The result showed that UV-B enhanced the interaction between GmILPA1 and GmGA2ox-like (Fig. 3h). However, UV-B treatment failed to affect the interaction between GmILPA1 and GmUBL1 or between GmGA2ox-like and GmUBL1 (Supplementary Fig. 7d–i). These results

demonstrated that UV-B light promotes the interaction between GmILPA1 and GmGA2ox-like.

## GmILPA1 mediates the degradation of GmGA2ox-like in a UV-B-dependent manner

The APC/C facilitates the ubiquitination and subsequent degradation of specific proteins by the UPS, raising the possibility that GmGA2ox-like might be targeted for degradation by GmILPA1. To test this hypothesis, we co-expressed constructs encoding N-terminally FLAG-tagged ubiquitin (FLAG-Ub) and C-terminally GFP-tagged GmGA2ox-like (GmGA2ox-like-GFP) in *N. benthamiana* leaves via Agrobacterium (*Agrobacterium tumefaciens*)–mediated transient expression, followed by Co-IP with anti-GFP antibodies. We detected a smear representing

**Fig. 3 | GmILPA1 interacts with GmGA2ox-like and GmUBL1. a** Yeast two-hybrid assays showing the interactions among GmILPA1, GmGA2ox-like, and GmUBL1. GmILPA1 and GmUBL1 were fused to GAL4-BD, and GmGA2ox-like was fused to GAL4-AD when encoded by the pGADT7 and pGBKT7 vectors. Serial dilutions of equivalent amounts of yeast were plated on synthetic defined (SD) medium without Leu (L), Trp (T) (SD-L-T), or without Leu (L), Trp (T), and His (H) (SD-L-T-H) containing 1 mM 3-amino-1,2,4-triazole (3-AT), with growth on triple dropout media indicating an interaction between the tested proteins. **b** A BiFC assay used to verify the interactions among GmILPA1, GmGA2ox-like, and GmUBL1 in vivo in *N. benthamiana* leaves. The non-interacting proteins, mGmGA2ox-like (mutated construct of D-Box motif) and GmPUB21 (a U-Box E3 Ubiquitin Ligases), were used as a negative control. Scale bars, 50 μm. **c–e** Co-IP of GmILPA1, GmGA2ox-like, and GmUBL1. Immunoprecipitation was performed with anti-GFP-agarose beads using *N. benthamiana* leaves co-infiltrated with the constructs *GmGA2ox-like-GFP* and *GmILPA1-MYC* (**c**), *GmUBL1-GFP* and *GmILPA1-MYC* (**d**), or *GmUBL1-GFP* and

*GmGA2ox-like-MYC* (**e**). **f** BiFC assays indicating that UV-B treatment promotes the interaction between GmILPA1 and GmGA2ox-like. *N. benthamiana* leaves were co-infiltrated with *GmGA2ox-like–nYFP* and *GmILPA1–cYFP* and exposed to 1 h of UV-B (21 μmol m$^{-2}$ s$^{-1}$) or kept in white light before imaging. Scale bars, 50 μm. **g** Relative YFP fluorescence intensity in the cytoplasm from the images in (**f**), (n = 30 cells, $P = 1.30 \times 10^{-5}$), the relative fluorescence intensities of cytoplasm and whole cells were quantified and the cytoplasm-to- background ratios are plotted. Data are presented as mean values ± SD, Student's *t*-test was used for the significance test, ***$P < 0.001$. **h** Co-IP assays showed that UV-B increased the interaction between GmILPA1 and GmGA2ox-like. *N. benthamiana* were co-transformed with *GmGA2ox-like-GFP* and *GmILPA1-MYC*, and treated with or without 12 h UV-B. Then, they were used in the co-immunoprecipitation assay. Immunoprecipitation was performed using GFP-Trap agarose beads, and the immunoblots were probed using anti-myc and anti-GFP antibodies. Source data are provided as a Source Data file.

poly-ubiquitinated GmGA2ox-like by immunoblotting with an anti-FLAG antibody (Fig. 4a), indicating that GmGA2ox-like is indeed subjected to polyubiquitination in plant cells. We then used a bioinformatics web tool to identify amino acids in GmGA2ox-like that are modified by ubiquitination, yielding the residues K394 and K407. We replaced each lysine residue by arginine (K394R, K407R, and K394R K2407R) and transiently expressed constructs encoding GmGA2ox-like-GFP harboring one or both mutations in *N. benthamiana* leaves to assess their effect on protein stability and ubiquitination. Accordingly, we immunoprecipitated total protein extracts with anti-GFP antibodies, followed by immunoblot analysis with anti-Ub antibodies. We observed smears in the immunoblots for intact GmGAox2-like and the K407R variant, but did not detect smears for the K394R or K394R K407R variant (Fig. 4b) which may indicate that K394 is the key residue for GmILPA1 ubiquitination.

To investigate whether GmILPA1 facilitates GmGA2ox-like degradation directly via ubiquitination, we performed in vitro ubiquitination assays. To this end, we used in vitro APC/C-ubiquitination assays, in which the APC/C complex (E3) immunoprecipitated from GmILPA1-GFP transgenic plants was incubated with Ub, ATP, E1, E2, and recombinant MBP-GmGA2ox-like. Our results showed that the E3 ligase GmILPA1 was auto-ubiquitinated as detected using the anti-Flag antibody (Fig. 4c). Indeed, GmILPA1 promoted the ubiquitination of GmGA2ox-like in vitro, as evidenced by the smear for GmGA2ox-like-MBP in immunoblots using anti-MBP antibodies; however, GmGA2ox-like$^{K394R}$ cannot be ubiquitinated (Fig. 4c).

We co-expressed the *GmGA2ox-like-GFP* and *GmILPA1-MYC* in *N. benthamiana* leaves, followed by Co-IP with anti-myc antibodies and immunoblotting with anti-ubiquitin antibodies. We detected a remarkably increased smear signal representing poly-ubiquitinated GmGA2ox-like in the presence of GmILPA1 (Fig. 4d). In addition, we used the same method to show that there was no remarkably increased smear signal of mGmGA2ox-like (mutated construct of D-Box motif) in the presence of GmILPA1 (Supplementary Fig. 9a). We interpreted this result as evidence that GmGA2ox-like can be ubiquitinated by GmILPA1.

As GmUBL1 has potential E3 ubiquitin ligase activity, we wondered if GmUBL1 might facilitate ubiquitination of GmGA2ox-like. We co-expressed the *GmGA2ox-like-GFP* and *GmUBL1-MYC* constructs in *N. benthamiana* leaves, followed by Co-IP with anti-myc antibodies and immunoblotting with anti-ubiquitin antibodies. Similarly, we detected a remarkably increased smear representing poly-ubiquitinated GmGA2ox-like in the presence of GmUBL1 (Fig. 4e). This result showed that GmGA2ox-like can be ubiquitinated by GmUBL1. To determine whether GmGA2ox-like can be ubiquitinated in soybean, we used a P62-agarose matrix that is capable of binding ubiquitinated proteins to enrich the ubiquitinated proteins from three independent transgenic seedlings stably expressing *35S:GmGA2ox-like-GFP* or from wild-type plants as a negative control. The bound proteins were used

for immunoblotting analysis with anti-GFP antibody. The ladder-like protein pattern was detected in the enriched proteins from three transgenic plants but not from WT plants (Fig. 4f). This result indicated that GmGA2ox-like can be ubiquitinated in soybean plants.

To understand how GmILPA1 regulates the protein stability of GmGA2ox-like in soybean plants, we examined the GmGA2ox-like protein levels by western blotting with anti-GA2ox-like antibody (see Supplementary Fig. 7b for GmGA2ox-like antibody specificity) in WT and *Gmilpa1-2* plants under both white light and white light supplemented with UV-B (600 μmol m$^{-2}$ s$^{-1}$ white light, 1.5 μmol m$^{-2}$ s$^{-1}$ UV-B, 25 °C, and a 12-h-light/12-h-dark photoperiod). We found lower levels of GmGA2ox-like under the white light supplemented with UV-B compared with that under the white light in the WT. In contrast, the GmGA2ox-like protein level was not reduced under the same conditions in the *Gmilpa1-2* plants (Fig. 4g, h). Moreover, V2 stage-grown WT and *Gmilpa1-2* mutant seedings under white light were transferred to white light supplemented with UV-B or continued to grow under white light for the indicated time periods, and our immunoblot data showed that there was no difference in the protein level of GmGA2ox-like between WT and *Gmilpa1-2* under white light (Fig. 4i, k). However, GmGA2ox-like showed substantial degradation in WT exposed to white light supplemented with UV-B from 10 h, but not in *Gmilpa1-2* under the same conditions (Fig. 4i, l). In addition, we performed cell-free degradation assays using total protein extracts from WT or *Gmilpa1-2* plants exposed to UV-B or maintained under white light and examined GmGA2ox-like abundance with anti-GmGA2ox-like. We did not observe degradation of GmGA2ox-like in either WT or *Gmilpa1-2* extracts in the absence of UV-B (Supplementary Fig. 9b, d). However, GmGA2ox-like showed substantial degradation in WT extracts exposed to UV-B from 60 min, (Supplementary Fig. 9c, e), but not in *Gmilpa1-2* extracts under the same conditions (Supplementary Fig. 9c, e). Incubation of MG132, an inhibitor of the 26 S proteasome degradation system, largely inhibited GmGA2ox-like degradation (Supplementary Fig. 9e). We concluded that GmILPA1 promotes GmGA2ox-like degradation by a 26 S proteasome in UV-B-dependent manner.

In light of the interaction of GmUBL1 with both GmILPA1 and GmGA2ox-like, we asked whether GmUBL1 played a role in GmGA2ox-like degradation with cell-free degradation assays using total protein extracts from *N. benthamiana* leaves transiently expressing *GmUBL1* or *GmGA2ox-like*. We determined that GmUBL1 can't promote the degradation of GmGA2ox-like directly, but requires GmILPA1 for this process (Fig. 4j, m). The result showed that the protein degradation is apparent from 90 min to 150 min, and these results indicated that GmUBL1 enhanced the GmILPA1-mediated degradation of GmGA2ox-like.

### *Gmilpa1-2* is sensitive to gibberellin and UV-B irradiation

The phytohormone GA contributes to internode elongation. We thus treated H12 and *Gmilpa1-2* soybean seedlings with 100 μmol GA$_3$ at the

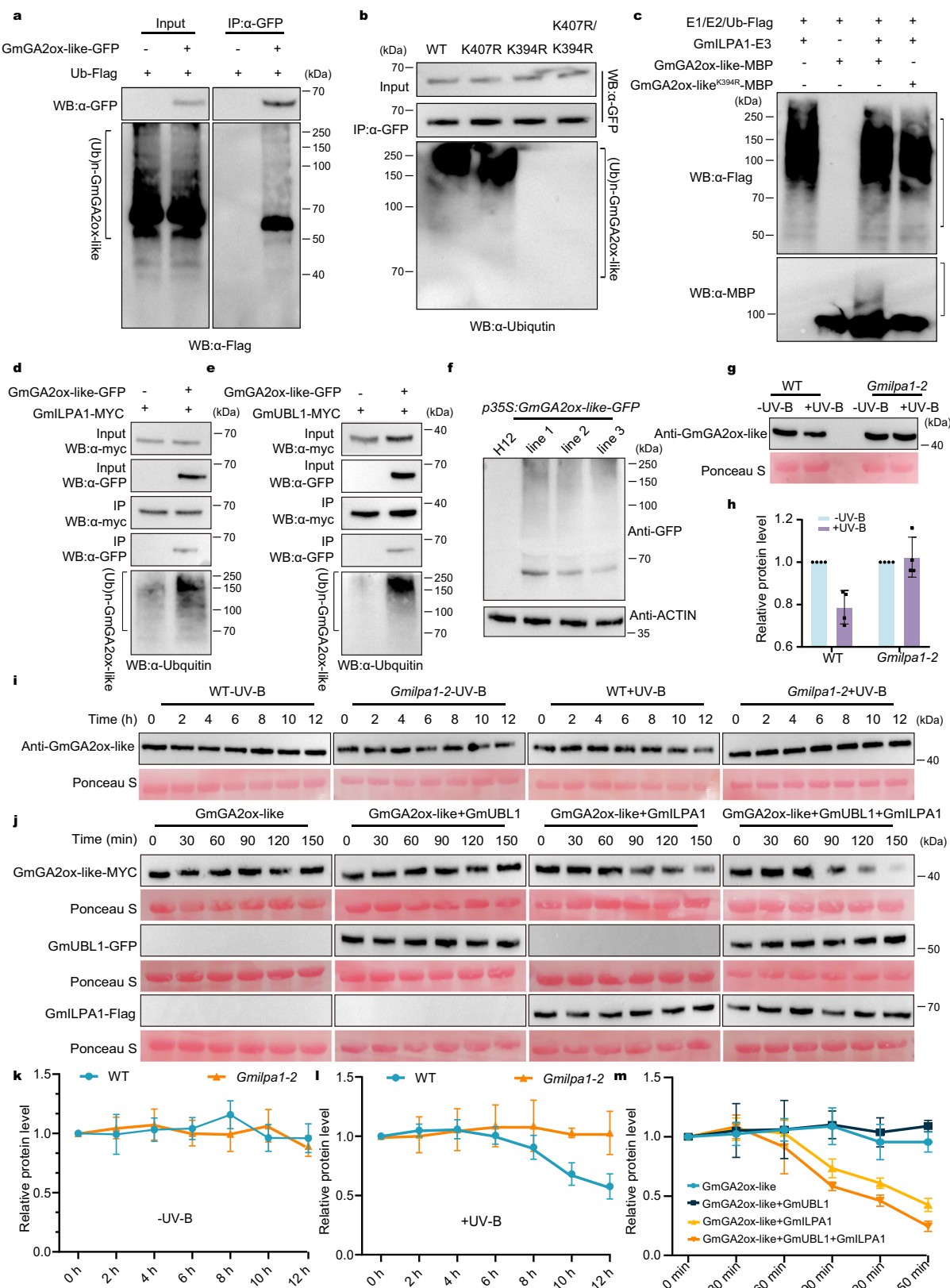

V1 stage (one unrolled trifoliate leaf) and measured plant height at the V4 stage (four unrolled trifoliate leaves). The results showed that exogenous GA₃ treatment restored *Gmilpa1-2* height to that of H12 when grown under sunlight (Fig. 5a, b). Treatment with the GA biosynthesis inhibitor paclobutrazol (PAC) efficiently inhibited elongation of H12, *Gmilpa1-2*, and *Gmilpa1* (Fig. 5c, d). Additionally, the exogenous

application of GA₃ rescued the PAC-induced growth deficiency of H12, *Gmilpa1-2*, and *Gmilpa1* seedlings and restored the dwarf phenotype of *Gmilpa1-2* and *Gmilpa1* treated with UV-B (Fig. 5c, d).

In addition, we measured the contents of endogenous GAs in H12 and *Gmilpa1-2* under white light and white light supplemented with UV-B, such as bioactive $GA_1$ and $GA_4$, as well as their precursors ($GA_{20}$

**Fig. 4 | GmGA2ox-like is ubiquitinated and degraded by the proteasome. a** In vivo ubiquitination of GmGA2ox-like as detected by co-immunoprecipitation (Co-IP) assay. **b** Identification of the ubiquitination sites essential for GmGA2ox-like ubiquitination. **c** Ubiquitination of GmGA2ox-like by GmILPA1 in vitro. Brackets denote the ubiquitinated bands. **d, e** Ubiquitination of GmGA2ox-like by GmILPA1 in *N. benthamiana* leaves via *Agrobacterium*-mediated transient expression. **f** GmGA2ox-like is ubiquitinated in plants. Ubiquitinated proteins were enriched from the P62-agarose matrix that was incubated with total proteins isolated from three independent transgenic plants stably expressing *GmGA2ox-like-GFP* or from H12 plants. **g** Regulation of GmGA2ox-like stability by *GmILPA1* in vivo. H12 and *Gmilpa1-2* seedlings grown under white light and white light supplemented with UV-B to the V2 stage, and GmGA2ox-like was checked with an anti-GmGA2ox-like antibody. The band intensities in western blots were quantified (**h**), Data are presented as mean values ± SEM, *n* = 3 independent experiments. **i** Immunoblots showing GmGA2ox-like protein levels in V2 stage seedlings of H12 and *Gmilpa1-2* grown in white light (600 μmol m$^{-2}$ s$^{-1}$, 12 h/12 h light/dark) and then transferred to white light supplemented with UV-B (1.5 μmol m$^{-2}$ s$^{-1}$, 12 h/12 h light/dark) for the indicated time periods. GmGA2ox-like abundance was detected with anti-GmGA2ox-like antibody and quantified (**k**, **l**). Data are presented as mean values ± SEM, *n* = 3 independent experiments. **j** GmGA2ox-like degradation in cell-free degradation assays in the presence of GmILPA1 and GmUBL1. *GmGA2ox-like-MYC*, *GmUBL1-GFP*, and *GmILPA1-FLAG* were individually infiltrated in *N. benthamiana*. *GmGA2ox-like-MYC* protein extracts were individually mixed with recombinant *GmUBL1-GFP*, *GmILPA1-FLAG*, *GmUBL1-GFP*, or *GmILPA1-FLAG*. Samples were collected at the indicated time points, and GmGA2ox-like abundance was probed by immunoblotting with anti-MYC antibody and quantified (**m**). Data are presented as mean values ± SEM, *n* = 3 independent experiments. Source data are provided as a Source Data file.

and GA$_9$) and catabolites (GA$_8$ and GA$_{34}$). In general, the endogenous GA content in both H12 and *Gmilpa1-2* plants showed a decreasing trend under white light supplemented with UV-B (Fig. 5e–j), which demonstrated that UV-B reduced the GA level in soybean. In addition, in consistent with the phenotype observed (Fig. 2c), there was no difference in the levels of GAs between H12 and *Gmilpa1-2* under white light, while the gibberellin GA$_1$, GA$_4$, GA$_{20}$ and GA$_9$ were more abundant in the WT plants than in *Gmilpa1-2* under white light supplemented with UV-B conditions, the non-bioactive GA$_8$ and GA$_{34}$ accumulated at a relatively high level in *Gmilpa1-2* relative to the H12 under same conditions (Fig. 5e–j), indicating the reduction rate of active GA in *Gmilpa1-2* was higher than that in H12. Therefore, our results demonstrated that GmILPA1 repressed the UV-B-induced decreasing of active GA levels to improve the UV-B tolerance in soybean.

RT-qPCR analysis of *GmILPA1* transcript levels in H12 seedlings treated with 100 μmol GA$_3$ and UV-B for 2, 4, 8, or 12 h indicated that *GmILPA1* was significantly upregulated compared to the control after GA$_3$ treatment (Supplementary Fig. 10a). However, relative *GmILPA1* transcript levels did not respond to UV-B treatment in H12 (Supplementary Fig. 10b). In addition, we observed that GmILPA1 accumulated within 12 h of treatment with GA$_3$ or UV-B light (Fig. 5m). The results showed that GA$_3$ and UV-B induced GmILPA1 protein accumulation.

In addition, RNA interference (RNAi) was used to silence GmGA2ox-like in *Gmilpa1-2*, which was designated as *Gmilpa1-2/GmGA2ox-like RNAi*. We grew H12, *Gmilpa1-2* and *Gmilpa1-2/GmGA2ox-like RNAi* seedlings under white light supplemented with UV-B, and the plant height of the T$_2$ generation of *Gmilpa1-2/GmGA2ox-like RNAi* seedlings were significantly higher than that of *Gmilpa1-2* (Fig. 5n, o, Supplementary Fig.10c), indicating that *Gmilpa1-2/GmGA2ox-like RNAi* notably rescued the *Gmilpa1-2* phenotype.

## Haplotypes 1, 2, 5 and 6 of *GmILPA1* were selected during domestication and modern breeding efforts

To understand the evolution of *GmILPA1* during domestication, we investigated the selection signals over the promoter and genomic regions of *GmILPA1* based on pairwise analyses of fixation index (FST), nucleotide diversity (π), and Tajima's D values among 444 individuals wild, landrace, and improved cultivars (Fig. 6a, Supplementary Fig. 11a, b). All three values across a 100-kb region containing *GmILPA1* revealed a clear selective sweep from the promoter and genomic regions of *GmILPA1* (Fig. 6a). This result indicated that GmILPA1 underwent an artificial selection during soybean subsequent genetic improvement. To investigate the natural variation at *GmILPA1* in germplasm resources, we performed a haplotype analysis across wild, landrace, and improved soybean populations[55]. We detected six main haplotypes, among which *Hap5* (76.5%) and *Hap1* (47.8%) mainly occurred in cultivated soybean, *Hap2* (60.5%) and *Hap6* (54.76%) were mainly present in landraces, and *Hap3* and *Hap4* were wild

soybean-specific haplotypes (Fig. 6b). To explore the evolution and spread of the six *GmILPA1* haplotypes during soybean domestication and breeding, we constructed its corresponding phylogenetic tree and haplotype network using 444 accessions. The analysis indicated that *Hap1–4* have a close genetic relationship, with *Hap3* perhaps representing the ancestral alleles from which all other haplotypes derived, and *Hap5* and *Hap6* are closely related to the wild soybean-specific haplotypes *Hap3* and *Hap4*, respectively (Fig. 6c, d).

We also noticed that the number of haplotypes decreased from six in wild soybeans to four in the landraces and improved cultivars (Fig. 6e–g), with the proportion of *Hap1 Hap2*, *Hap5* and *Hap6* increasing from wild soybeans to the landraces, the proportion of *Hap1* and *Hap5* gradually increasing from wild soybeans to the landraces and to improved cultivars. The relative increase in *Hap5* frequency was more substantial than the other haplotypes (Fig. 6e–g), suggesting that mainly *Hap1* and *Hap5*, especially *Hap5*, have been strongly selected for by humans during soybean domestication and modern breeding efforts.

## An INDEL in the *GmILPA1* promoter confers variation of plant height in response to UV-B in cultivars

Sequence analysis using the Plant-CARE database identified six *Haps*. Of them, *Hap5* was the only haplotype with an INDEL (deletion of 13 bp) at position 1,877,295 bp (Indel-665) of the *GmILPA1* promoter, which contains a light-responsive *cis*-regulatory elements (CREs). The deletion of this CRE may affect the transcriptional activity of GmILPA1 under UV-B conditions.

We wished to investigate *GmILPA1* transcriptional activity under UV-B stress. Accordingly, we cloned *GmILPA1* promoters from *Hap1* to *Hap6*, and constructed their vectors *GmILPA1pro^Hap1^:GUS* to *GmILPA1pro^Hap6^:GUS*, and the *GmILPA1^Hap5^* promoter with indel vector *GmILPA1pro^Hap5/indel^:GUS*. We transiently expressed each construct in *N. benthamiana* leaves via *Agrobacterium*-mediated infiltration to test their transcriptional output with and without UV-B treatment. Histochemical staining showed no difference in GUS signal among *GmILPA1pro^Hap1^:GUS*, *GmILPA1pro^Hap3^:GUS*, *GmILPA1pro^Hap5^:GUS*, *GmILPA1pro^Hap5/indel^:GUS* and *GmILPA1pro^Hap6^:GUS* when infiltrated leaves were maintained under white light (Fig. 7a). We also did not observe a difference in GUS signal for *GmILPA1pro^Hap5^:GUS* regardless of UV-B exposure (Fig. 7a). However, the GUS signal for *GmILPA1pro^Hap1^:GUS*, *GmILPA1pro^Hap3^:GUS*, *GmILPA1pro^Hap5/indel^:GUS* and *GmILPA1pro^Hap6^:GUS* appeared higher than that of *GmILPA1pro^Hap5^:GUS* when the infiltrated leaves were exposed to UV-B (Fig. 7a). We confirmed these results with a quantitative GUS assay (Fig. 7b). We performed RT-qPCR on the gene *bar* on the GUS vector as infiltration controls (Supplementary Fig. 12a). These results suggested that the differential transcriptional activities between *Hap5* and other haplotypes under UV-B expose is caused by the deletion of CRE in the *GmILPA1* promoter of *Hap5*, which has a significant effect on the response to UV-B.

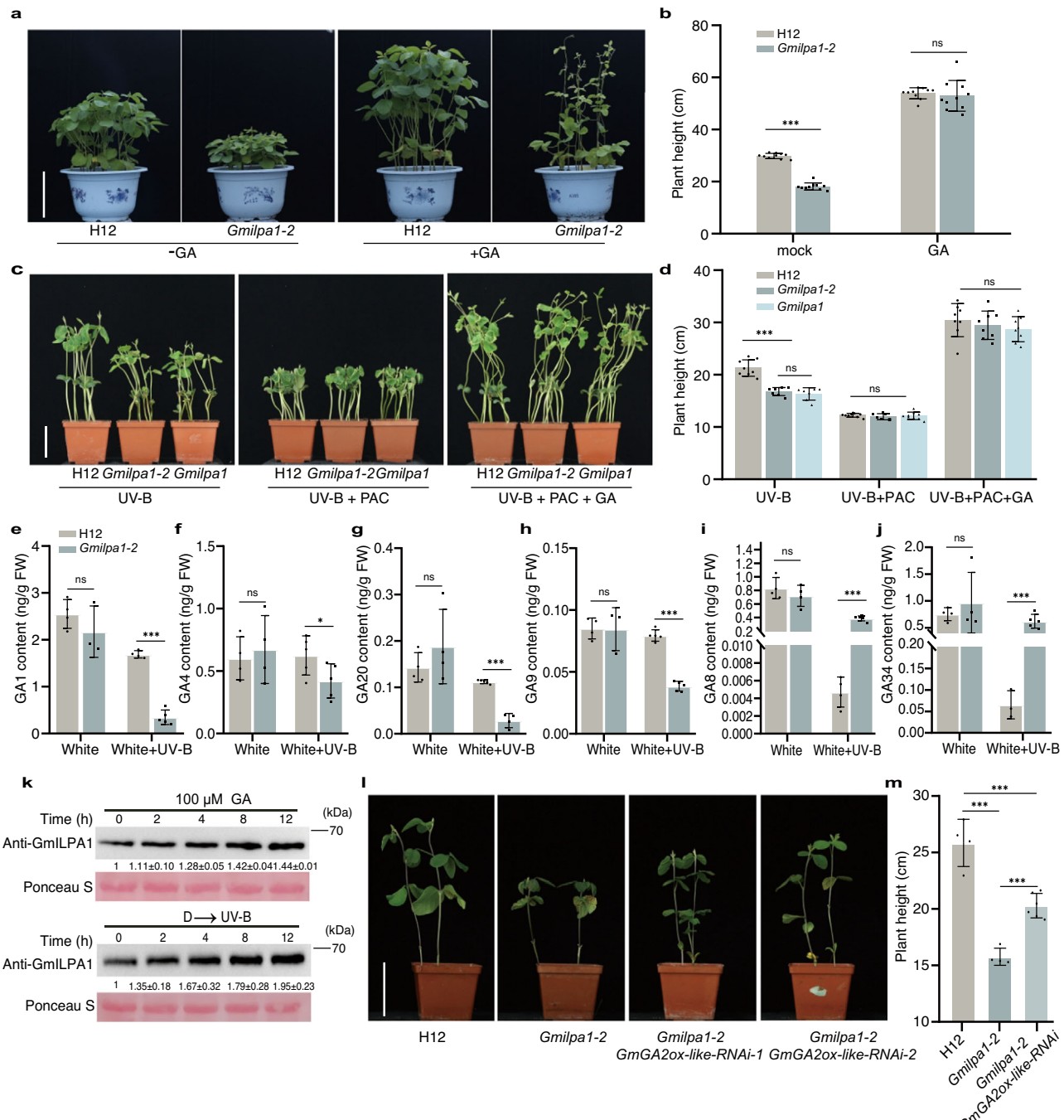

**Fig. 5 | Exogenous gibberellin application can rescue the dwarf phenotype of** ***Gmilpa1-2.*** **a** Phenotypes of Hedou 12 (H12) and *Gmilpa1-2* seedlings grown under sunlight treated with GA₃ at the V1 stage (one unrolled trifoliate leaf), Scale bar, 20 cm. **b** Plant height of the seedlings at the V4 stage (three unrolled trifoliate leaves) in (**a**), ($n = 10$ biologically independent samples; $P = 1.92 \times 10^{-14}$ under mock condition). **c** Phenotypes of H12, *Gmilpa1-2*, and *Gmilpa1* seedlings treated with PAC alone and PAC with GA₃ treated with UV-B for 14 days, scale bar, 10 cm. **d** Plant height of the seedlings in (**c**), ($n = 7$ biologically independent samples; $P = 1.12 \times 10^{-5}$ under UV-B condition). **e**–**j** Endogenous GAs levels in H12 and *Gmilpa1-2*. GA levels were measured using apical buds of V2-stage (two unrolled trifoliate leaves) seedlings grown under white light and white light with UV-B, ($n \geq 3$ biologically independent samples; $P = 6 \times 10^{-4}$ in **e** $P = 0.0134$ in **f** $P = 6.90 \times 10^{-6}$ in **g** $P = 3.71 \times 10^{-7}$ in **h** $P = 1.29 \times 10^{-7}$ in **i** $P = 4 \times 10^{-4}$ in **i**). *P*-values were calculated compared to H12 seedlings by two-tailed Student's *t*-tests; ***$P < 0.001$, *$P < 0.05$; ns not

significant. **k** Immunoblots showing GmILPA1 abundance in seedlings of H12 at the V1 stage (unrolled one trifoliate leaf) treated with 100 μM GA₃ and UV-B for the indicated times. **l** Phenotypes of H12, *Gmilpa1-2*, and T₂ progeny lines of *Gmilpa1-2/GmGA2ox-like RNAi* grown under white light with UV-B (600 μmol m⁻² s⁻¹, 1.5 μmol m⁻² s⁻¹ UV-B, 12-h light/12-h dark). Scale bar, 10 cm. **m** Plant height of the seedlings, ($n = 4$ biologically independent samples for H12 and *Gmilpa1-2*, $n = 6$ biologically independent samples for *Gmilpa1-2/GmGA2ox-like-RNAi*, $P = 1.01 \times 10^{-4}$ in H12 and *Gmilpa1-2*, $P = 5.21 \times 10^{-4}$ in H12 and *Gmilpa1-2/GmGA2ox-like-RNAi*, $P = 9.29 \times 10^{-5}$ in *Gmilpa1-2* and *Gmilpa1-2/GmGA2ox-like-RNAi*). In **b**, **d**, **m**, Data are presented as mean values ± SD, *P*-value is calculated with a one-way ANOVA analysis–Tukey comparison, and the columns labeled without the same alphabet are significantly different ($P < 0.05$, two-sided). ***$P < 0.001$, ns, not significant. Source data are provided as a Source Data file.

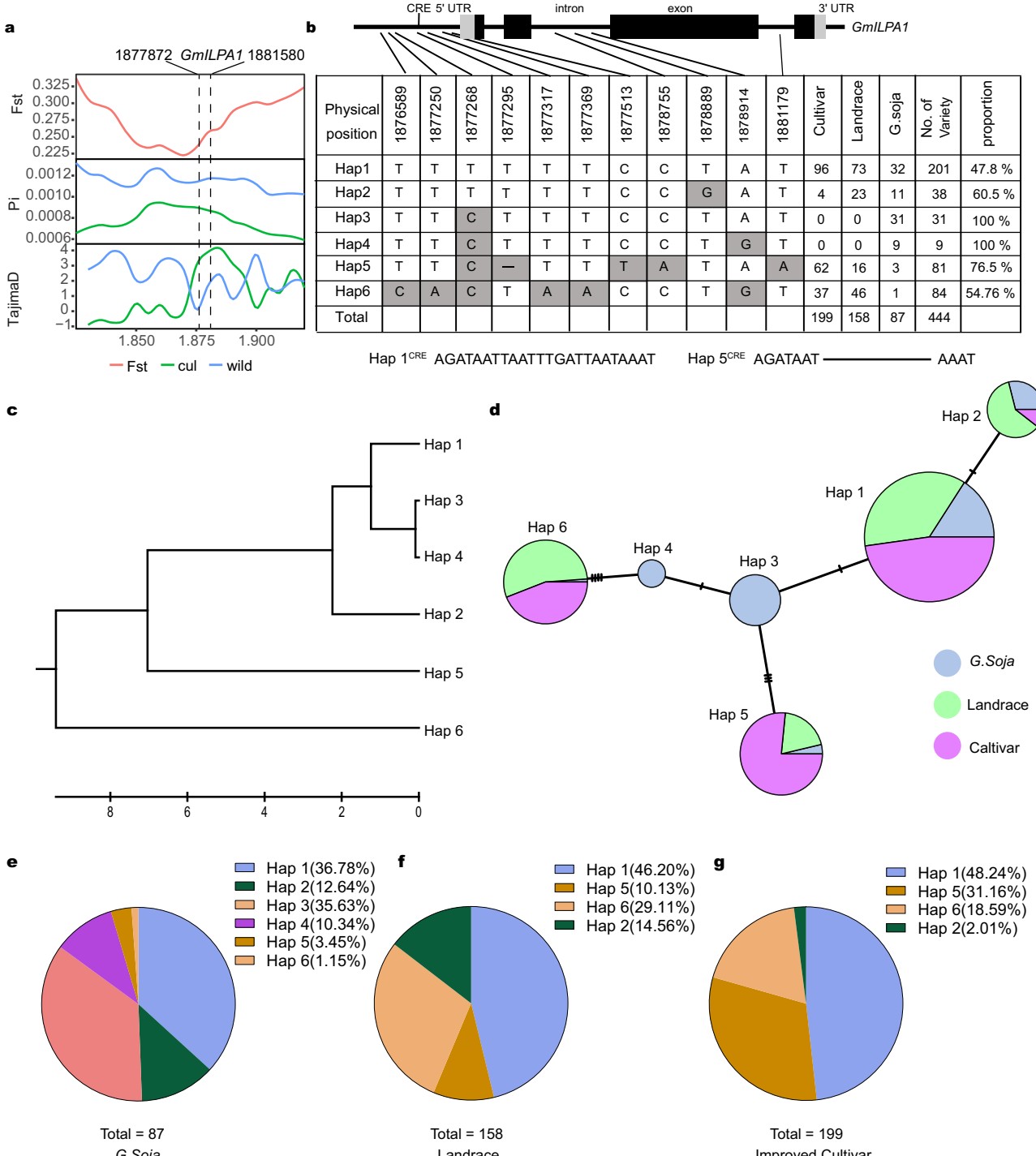

**Fig. 6 | Haplotype analysis and origin of *GmILPA1* in soybean accessions from different populations of China. a** FST, nucleotide diversity, and Tajima's D values over the genomic region containing the *GmILPA1* locus (-100 kb) between wild and cultivar soybean germplasms. **b** *GmILPA1* haplotypes in natural populations. Top, schematic diagram of the *GmILPA1* gene structure; gray, untranslated regions (UTRs); black, exons; black lines, promoter and introns. The CRE in the promoter region represents a light-responsive *cis*-regulatory element; –, 13-bp deletion. Bottom, *GmILPA1* polymorphism across accessions relative to the Williams 82

reference genome (*Hap1*). The number of varieties for each haplotype (*Hap1–6*) is shown to the right. The cultivated soybean Hedou 12 belongs to *Hap5*.
**c** Evolutionary relationship of the *GmILPA1* six haplotypes. **d** Haplotype origins of *GmILPA1* as shown by the median-joining method. Circle size is proportional to the number of accessions, while circle colors represent the different soybean groups: blue, wild soybean (*S. soja*); green, landraces; magenta, cultivars. **e**–**g** Pie charts representing the distribution of each haplotype in wild soybeans, landraces, and improved cultivars.

Modern breeding programs can progressively accumulate elite alleles. To further clarify whether accessions with improved cultivars differ from each other when grown under natural conditions, we measured their height. Haplotypes harboring *Hap1*, *Hap2* and *Hap6* were taller than those with *Hap5* (Fig. 7c). We also investigated other

agronomic traits in accessions carrying different haplotypes, which revealed that *Hap5* generally have fewer pods, while hundred-seed weight and grain weight per plant were significantly higher than in other haplotypes (Fig. 7d–f). These results suggested the high yield potential of *Hap5* for soybean breeding programs.

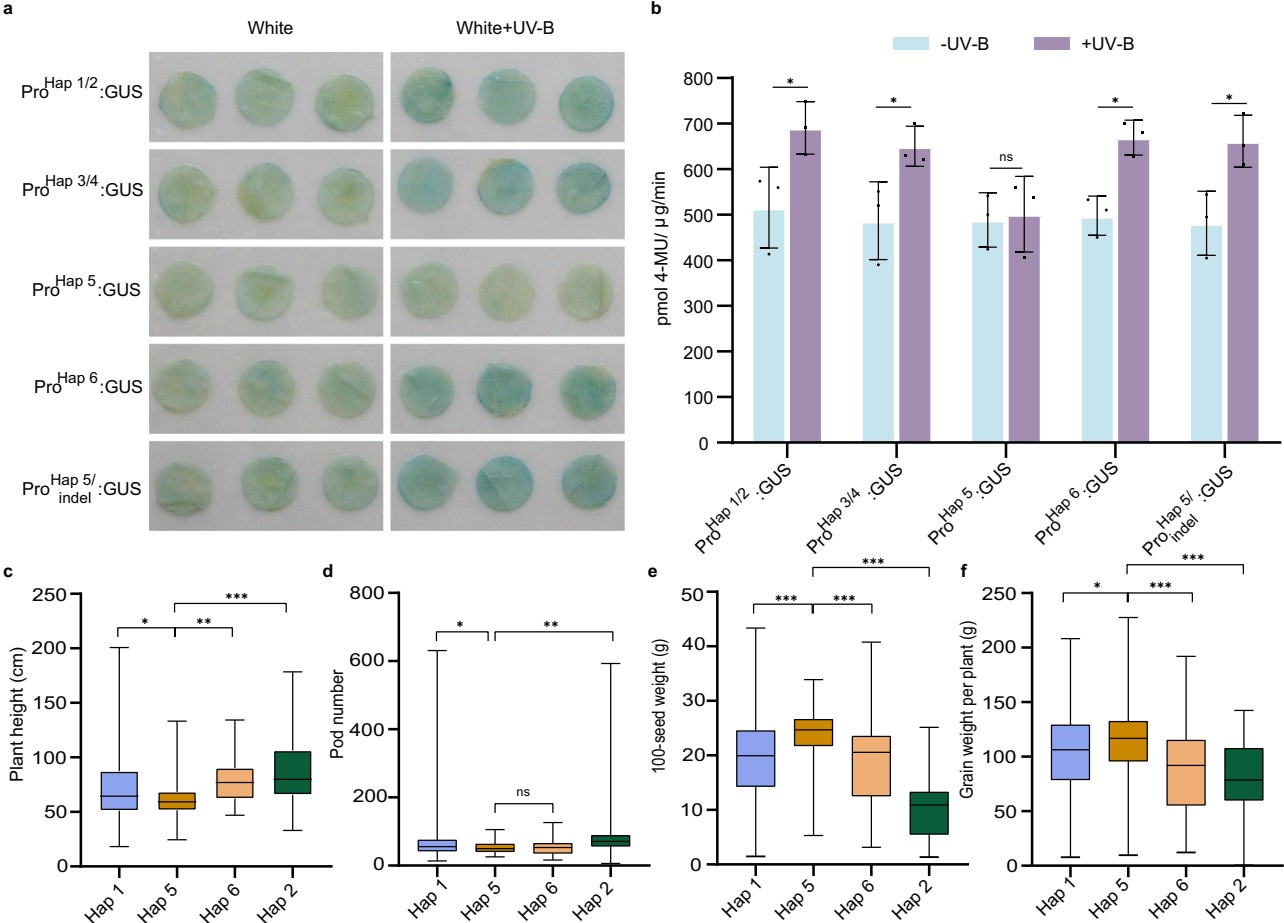

**Fig. 7 | Different Haplotypes of *GmILPA1* confers variation of plant height in response to UV-B in cultivars. a** GUS activity detected by histochemical staining in *Nicotiana benthamiana* leaves transiently expressing the *GUS* reporter gene under the control of *GmILPA1Hap1*- *GmILPA1Hap6* and the *GmILPA1Hap5* promoter with indel. For UV-B treatment, *N. benthamiana* were treated with UV-B (1.5 µmol m⁻² s⁻¹) for 12 h before histochemical staining. **b** GUS activity from *N. benthamiana* leaves transiently expressing *GUS* under the control of *GmILPA1Hap1*- *GmILPA1Hap6* and the *GmILPA1Hap5* promoter, as measured by fluorometric assay and expressed as pmol 4-methylumbelliferone µg⁻¹ protein min⁻¹, (*n* = 3 independent experiments, *P* = 0.0457 in ProHap1/2: GUS, *P* = 0.0425 in ProHap3/4: GUS, *P* = 0.0413 in ProHap6: GUS and *P* = 0.0263 in ProHap5/indel: GUS). Data are presented as mean values ± SD, Student's *t*-test was used for the significance test, ns, not significant, *P < 0.05. **c**–**f** Summary of major agronomic traits plants with the *GmILPA1Hap1*, *GmILPA1Hap2*,

*GmILPA1Hap5* or *GmILPA1Hap6* allele. (*P* = 0.0409 in Hap 1 and Hap 5, *P* = 0.0088 in Hap 6 and Hap 5, *P* = 0.0006 in Hap 2 and Hap 5) (**c**), (*P* = 0.0233 in Hap 1 and Hap 5, *P* = 0.0017 in Hap 2 and Hap 5) (**d**), (*P* = 2.61 × 10⁻⁶ in Hap 1 and Hap 5, *P* = 3.57 × 10⁻⁶ in Hap 6 and Hap 5, *P* = 8.95 × 10⁻²¹ in Hap 2 and Hap 5) (**e**), (*P* = 0.0143 in Hap 1 and Hap 5, *P* = 5.35 × 10⁻⁶ in Hap 6 and Hap 5, *P* = 1.33 × 10⁻⁶ in Hap 2 and Hap 5) (**f**). *n* = 179 for *GmILPA1Hap1*, *n* = 33 for *GmILPA1Hap2*, *n* = 76 for *GmILPA1Hap5*, *n* = 81 for *GmILPA1Hap6*. The box indicates the range from lower to upper quartiles, and the bar ranges the minimum to maximum observations; the significance of the difference is calculated by comparison with Hap 5 with a one-way ANOVA analysis–Tukey comparison and the columns labeled without the same alphabet are significantly different (P < 0.05, two-sided). *P < 0.05, **P < 0.01, ***P < 0.001; ns, not significant. Source data are provided as a Source Data file.

## Discussion

UV-B irradiation is an environmental stress that can inhibit the elongation of hypocotyls and stems, resulting in dwarf plants[56]. The regulation of hypocotyl elongation in response to UV-B light is usually mediated by its dedicated photoreceptor UVR8[32,33,35,57] and the E3 ligase CONSTITUTIVELY PHOTOMORPHOGENIC1 (COP1), which binds to UVR8 monomers to initiate UV-B signaling[57,58]. However, whether other E3 ligases are involved in plant height in response to UV-B is unknown. In this study, we identified a subunit of the APC/C in soybean, GmILPA1, as an APC8-like protein with E3 ubiquitin ligase activity. We also showed that UV-B can promote the accumulation of GmILPA1 to control plant height of soybean under UV-B exposure (Figs. 2a, c and 5k). Thus, GmILPA1 might be involved in UV-B acclimation in soybean.

So far, little is known about plant targets of the APC/C compared to their counterparts in animals and yeast (*Saccharomyces cerevisiae*). Known substrates of the APC/C in plants include cyclin (CYCA3;1/ CYCA2;3) and CYCB2;1[59–61], the double-stranded (ds) RNA-Binding

Protein 4 (DRB4)[62], a member of the plant-specific GRAS family MONOCULM1 (MOC1)[63], and a small GTP binding protein ROOT ARCHITECTURE-ASSOCIATED1 (RAA1)[64–66]. In this study, we identified GmGA2ox-like as an interacting protein of GmILPA1 in a yeast two-hybrid screen (Fig. 3a). GmGA2ox-like harbors the typical D-box (the destruction box, RXXLXXXN) that is recognized and degraded by the APC/C (Supplementary Fig. 6f). Further we found that GmGA2ox-like was recognized by GmILPA1 (APC/C) and degraded only upon UV-B exposure in soybean (Fig. 4i). Therefore, we indicated that GmGA2ox-like is a novel substrate of the APC/C.

A previous study showed that UV-B-induced morphological adjustments are associated with the dynamic stability in phytohormones like GA[37]. UV-B induces the expression of the GA catabolic genes *GA2ox2* and *GA2ox8* in *Arabidopsis*, thus inhibiting hypocotyl elongation, and UV-B influences the expression of GA metabolic genes requiring UVR8[67,68]. However, whether GA2ox function is also modulated at the post-transcriptional level has remained unclear. In this study, we found that GmGA2ox-like is recognized by the APC/C and

degraded only upon UV-B exposure in soybean (Fig. 4i), and indicated that the degradation of GmGA2ox-like by GmILPA1 is UV-B dependent. Moreover, we found that another E3 ligase GmUBL1 also interacted with GmGA2ox-like and ubiquitinated GmGA2ox-like, which further promoted the degradation of GmGA2ox-like (Fig. 4j). We speculated that the enhanced ubiquitination of GmGA2ox-like further accelerates its degradation. Furthermore, the levels of active GAs in the *Gmilpa1-2* mutant were significantly lower than in the WT (Fig. 5e, f), and exogenous GA applications restored the height of the *Gmilpa1-2* mutant to that of the WT under UV-B light (Fig. 5c). In agreement with these results, overexpression of *GmGA2ox-like* led to dwarfism in soybean (Supplementary Fig. 6d), similar to the *Gmilpa1-2* mutant. By measuring the GAs contents in WT and *Gmilpa1-2* plants, we found that UV-B-induced reduction rate of active GA in *Gmilpa1-2* was higher than that in H12 (Fig. 5e–j). Therefore, we demonstrated that GmILPA1-GmGA2ox-like module enhances the UV-B tolerance to maintain growth in soybean by repressing UV-B-induced reduction of active GA levels. Taken together, these findings improve our understanding of the mechanisms that how soybean plants improve its tolerance to UV-B stress. Interestingly, exogenous GA only restored the plant height of *Gmilpa1-2*, while *Gmilpa1-2* plants was different with wild type for the overall morphology (Fig. 5a). Considering the importance of hormone homeostasis, we hypothesize that this phenomenon might be GA-dose dependent. Moreover, previous study reported that GA treatment did not restore the petiole angle of the *Gmssp* (*Gmilpa1*) mutant[69]. Therefore, in addition to mediating plant height via GA metabolism, *GmILPA1* might be involved in different growth and development processes, such as leaf petiole angle regulation, through other mechanisms.

Although E3 ligase itself has ubiquitination ligase activity and is able to ubiquitinate substrates, it may also be regulated by other proteins. For example, SPA can interact with COP1 and enhance the ubiquitination activity of COP1 in vitro[70], while COP1 can be ubiquitinated and degraded mediated by CSU1 (a RING-finger E3 ubiquitin ligase) in darkness[71]. In this study, we identified another E3 ligase GmUBL1 as an interacting protein of GmILPA1 and GmGA2ox-like (Fig. 3a). GmUBL1 could induce the ubiquitination of GmGA2ox-like but did not lead to degradation of GmGA2ox-like (Fig. 4j). Further studies demonstrated that GmUBL1 promoted the GmILPA1-mediated degradation of GmGA2ox-like, probably due to enhanced ubiquitination (Fig. 4j). Therefore, we speculate that GmUBL1 might enhance the ubiquitin E3 ligase activity of GmILPA1, suggesting a more complicated regulation mechanism for GmILPA1 activity and GmGA2ox-like stability.

Achieving ideal plant architecture is an important objective of artificial selection and modern breeding, as it can increase grain yield by allowing denser planting and raising photosynthetic efficiency[72]. Due to artificial selection of specific alleles (haplotypes) during crop domestication and breeding, plants with new traits meeting human needs were produced, but this is often accompanied by a loss of allele diversity[73]. In this study, we detected evidence of artificial selection over the *GmILPA1* locus during soybean domestication and modern genetic improvement. We showed that *Hap1* and *Hap5* were the main selected haplotypes in modern cultivated germplasms, while *Hap2* and *Hap6* gradually declined in frequency from wild soybeans to modern cultivated germplasms (Fig. 6c–e). This pattern suggested that *Hap1* and *Hap5* are haplotypes that have mainly undergone artificial selection.

Mutations in exons that lead to loss of function often cause strong phenotypic changes, but alterations in transcription levels are also an important genetic basis for trait transition[74]. The selection of *Hap5* in modern improved cultivars may be associated with the light-responsive CRE present in *Hap1*, *Hap2* and *Hap6*, but not in *Hap5*; notably, relative *GmILPA1* transcript activity was higher harboring *Hap1*, *Hap2*, and *Hap6* compared with those with *Hap5* upon exposure

to UV-B in the *N. benthamiana* transient expression system (Fig. 7a, b), indicating that the deletion of the light-responsive *cis*-regulatory element CRE may affect the response to UV-B. Further we also investigated plant height in accessions carrying different haplotypes, and found that *Hap5* with the deletion of CRE generally is shorter than other haplotypes (Fig. 7a). Together, these results suggested the INDEL in the *GmILPA1* promoter may be involved in this phenotypic difference.

Given that northern China has more ultraviolet radiation than southern China[75], we surveyed the geographical distribution of *Hap1*, *Hap5* and *Hap6* using soybean cultivars collected from 7 geographical regions in northern China (Jilin, Liaoning and Heilongjiang provinces) and southern China (Fujian, Guizhou, Jiangxi and Hunan provinces). The frequency of *Hap1* and *Hap6* increased from the north region to the south region. By contrast, *Hap5* only appears in the north of China (Fig. 8a), suggesting that the growth of *Hap5* is more suitable for high UV-B environments. Furthermore, we analyzed *GmILPA1* transcript levels after UV-B treatment in 30 randomly selected soybean accessions each for *Hap1* and *Hap5*. We observed that *GmILPA1* from *Hap1* germplasm is more highly expressed than in *Hap5* germplasm in response to UV-B light (Supplementary Fig. 12b). In addition, we measured plant height for all 60 accessions grown under white light only or exposed to UV-B light. Notably, plant height for *Hap1* germplasms was significantly higher than that of *Hap5* germplasms upon exposure to UV-B light (Supplementary Fig. 12c). Therefore, we speculated that the deletion of CRE in *GmILPA1* promoter of *Hap5* leads to the reduction of plant height under UV-B radiation, which is a manifestation of the soybean's own protection resistant to UV-B radiation (Fig. 8b). Taken together, *Hap5* variant could be used to develop new soybean cultivars with ideotype and tolerant to UV-B radiation. Importantly, *GmILPA1* is held to be a promising potential target for single base editing aimed at raising the planting density and lodging-resistant of soybean in the future.

## Methods
### Plant materials and growth conditions
The *Gmuid1* (also named *Gmilpa1-2*) mutant was generated in our laboratory; *Gmilpa1* was provided by Xianzhong Feng. Both mutants are in the soybean (*Glycine max* L.) Heidou 12 background. To investigate the phenotype of Hedou 12 and *Gmuid1* under natural sunlight (70–100 μW cm$^{-2}$ UV-B), plants were grown at Qingdao, Shandong (35.35° N, 119.3° E) during the summer. The UV intensity was measured using a 254 UV light meter (manufactured by Beijing Normal University). For laboratory experiments, plants were cultivated in a glasshouse (with no UV-B) under natural sunlight and in a phytotron equipped with LED lights. For treatments with white light, Hedou 12 and the *Gmuid1* mutant were grown in a growth chamber (600 μmol m$^{-2}$ s$^{-1}$ white light, no UV-B, 25 °C, 12-h-light/12-h-dark photoperiod). For UV-B treatment, weak narrow-band UV-B coupled with white light was used (600 μmol m$^{-2}$ s$^{-1}$ white light, 1.5 μmol m$^{-2}$ s$^{-1}$; TL20W/01RS tubes, Philips; 12-h-light/12-h-dark photoperiod). For treatment with gibberellin, the leaves of V1-stage plants were sprayed with 100 μM GA$_3$ (Sigma-Aldrich, USA) using a hand-held aerosol-propelled sprayer, spray plants with water as negative control

### Map-based cloning of *GmILPA1* (*GmUID1*)
For map-based cloning of *GmILPA1*, an F$_2$ population was generated by crossing the *Gmuid1* mutant in the Heidou 12 background and Williams 82. F2 plants with the mutant phenotype were used for mapping with the markers listed in Supplementary Data 2. For bulk segregant analysis, 80 plants (40 with a wild-type phenotype and 40 with the mutant phenotype) were selected from the F$_2$ population. Genomic DNA was extracted from all individual plants and mixed in equal amounts to generate the wild-type pool (W-pool) and the mutant pool (M-pool). Sequencing libraries were constructed for each pool and the parents

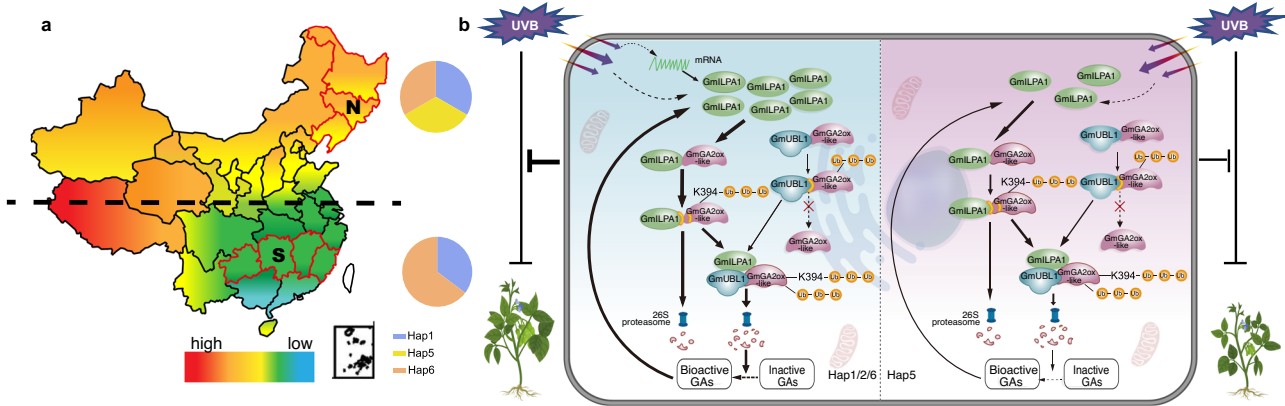

**Fig. 8 | A working model of the regulation of plant height by the GmILPA1-GmUBL1-GmGA2ox-like module in soybean. a** Geographical distribution of *Hap1*, *Hap5* and *Hap6*. N, north of China, S, south of China. The pie chart represents their frequency. **b** The working model of GmILPA1 regulating of plant height in *Hap1/2/6* and *Hap5*. In *Hap1/2/6*, under the UV-B irradiation, GmILPA1 accumulation is induced at transcriptional and protein levels. GmGA2ox-like is ubiquitinated by GmILPA1 and GmUBL1, and GmILPA1 mediates the degradation of GmGA2ox-like, GmUBL1 enhanced the degradation of GmGA2ox-like. As a result, the increase of bioactive GAs content promoted the growth of soybean. In *Hap5*, under the UV-B irradiation, GmILPA1 accumulation is induced only at protein level, eventually leads to shorter plant height in *Hap5*. GA3 induces GmILPA1 protein accumulation both in *Hap1/2/6* and *Hap5*.

Heidou 12 and Williams 82. Genomic DNA was sheared for preparation of the sequencing library according to the standard protocol of the Illumina TruSeq DNA PCR-free prep kit. Reads of the two bulks and the parental line were aligned to the reference genome using bwa software with default parameters; SNP calling was performed using GATK software. SNPs with a sequencing depth of <5 were filtered. Then, the SNP index was calculated, and SNPs with a SNP index of <0.3 in both bulks were filtered to reduced false positive detection of SNPs. Statistical confidence intervals of Δ (SNP index) were calculated under the null hypothesis of no quantitative trait loci (QTLs) following procedures reported previously[76]. The primers are listed in Supplementary Data 2.

### Plasmid construction and transformation
The coding sequences of *GmILPA1* and *GmGA2ox-like* were amplified from Hedou 12 complementary DNA (cDNA) and cloned in-frame with the *GFP* coding sequence into the pB7FWG2 vector under the control of the cauliflower mosaic virus (CaMV) 35S promoter. The resulting constructs were introduced into the *Gmilpa1-2* mutants and Hedou 12 by Agrobacterium (*A. tumefaciens*)–mediated transformation. The primers are listed in Supplementary Data 2.

### RNA extraction and RT-qPCR
Total RNA from examined organs was isolated using Trizol reagent (Invitrogen). First-strand cDNA was synthesized from 1 μg of total RNA using a MonScript RTIII All-in-One Mix with dsDNase (Monad). For qPCR, each sample was subjected to three technical replicates, using a 2× M5 HiPer Realtime PCR Super Mix on an Applied Biosystems instrument. The level of *GmELF1b* was used as the internal control[77].

### GA measurements
To quantify the contents of GAs, H12 and *Gmilpa1-2* seedlings were grown under white light and white light with UV-B. The stem apexes of the seedlings were collected, frozen in liquid nitrogen, ground to a fine powder, and extracted with the traction method (methanol/water/formic acid = 15:4:1, V/V/V). The extracts were vortexed and centrifuged at 4694 × *g* under 4 °C for 10 min. The supernatants were dried by evaporation under the flow of nitrogen gas at room temperature, then dissolved in 200 μl of methanol. The sample extracts were analyzed using an LC-ESI-MS/MS system (HPLC, Shimpack UFLC SHIMADZU CBM30A system; MS, Applied Biosystems 6500 Triple), and the data were analyzed by Metware

Biotechnology Co., Ltd (Wuhan, China). At least three replicates of each assay were performed, the information of ions monitored was provided in Supplementary Data 1.

### Immunoblotting
Total proteins were extracted with extraction buffer (50 mM Tris-HCl, pH 7.5, 150 mM NaCl, 5 mM EDTA, 2 mM DTT, 10% [v/v] glycerol, 1 mM PMSF, 1% [v/v] PVPP, and 1× protease inhibitor cocktail). Samples were immediately boiled for 10 min and then centrifuged at 14,000 × *g* for 10 min at room temperature. Proteins from the supernatant were used for immunoblotting. The anti-GmILPA1 and anti-GmGA2ox-like antibodies were produced by MW BIOTECH (HK) LIMITED.

### Yeast two-hybrid and bimolecular fluorescence complementation (BiFC) assays
The full-length coding sequences of *GmILPA1*, *GmGA2ox-like*, and *GmUBL1* were cloned into pGADT7 or pGBKT7 vectors to generate the bait and prey constructs, respectively. The appropriate pairs of constructs were co-transformed into yeast strain Y2HGold and grown on selective plates at 30 °C for 2–4 days. For BiFC assays, the same coding sequences were cloned in-frame with the *nYFP* or *cYFP* sequences to generate *GmILPA1-cYFP*, *GmGA2ox-like-nYFP*, *GmGA2ox-like-cYFP*, and *GmUBL1-nYFP*. The resulting constructs were transformed into Agrobacterium strain GV3101, and the appropriate pairs were co-infiltrated into *N. benthamiana* leaves together with the silencing suppressor P19. Fluorescent signals were visualized by using the LSM-700 laser scanning confocal microscope (Zeiss) and the signal intensities of YFP were determined by ImageJ software. The primers used in the generation of the relevant construct are listed in Supplementary Data 2.

### Microscopy procedures
To observe longitudinal parenchyma cells of H12 and *Gmuid1*, the fifth internode was fixed in FAA solution (5% formaldehyde, 50% absolute alcohol, 5% acetic acid) and then embedded in paraffin using a Histo-Core Arcadia machine (Leica Biosystems). Sections (7 μm in thickness) were obtained using a HistoCore MULTICUT (Leica Biosystems) and observed and photographed using a microscope (Olympus BX51) with a DP72 camera (Olympus). For the YFP fluorescence signals, fluorescence microscopy of abaxial leaf epidermal cells was performed using a confocal laser scanning microscope (ZEISS LSM 700; ZEISS, Oberkochen, Germany).

## Co-immunoprecipitation (Co-IP) assays

The CDSs of *GmGA2ox-like* and *GmUBL1-GFP* were cloned into the modified vector to generate *GmGA2ox-like-GFP* and *GmUBL1-GFP*. The CDS of *GmILPA1* and *GmGA2ox-like* were cloned into the modified vector to generate *GmILPA11-myc* and *GmGA2ox-like-myc*. The *GmILPA1-myc*, *GmGA2ox-like-GFP*, *GmUBL1-GFP*, or *GmGA2ox-like-myc* construct was transformed into Agrobacterium strain GV3101. *Agrobacteria* harboring the indicated constructs were co-infiltrated into the epidermal cells of *N. benthamiana* leaves. 100 μM MG132 was co-injected with the *Agrobacterium* suspension to prevent the degradation of GmGA2ox-like-GFP protein. The plants were grown for 3 days at 25 °C before the infiltrated leaves were collected. Total proteins were extracted in extraction buffer containing 20 mM HEPES, pH 7.4, 150 mM NaCl, 1 mM EDTA, 0.5% (v/v) Triton X-100, 1 mM PMSF, 10% (v/v) glycerol, 1× protease inhibitor cocktail, and 0.1% (v/v) β-mercaptoethanol. After centrifugation for 15 min at 14,000 × $g$ at 4 °C. The supernatant was mixed with 15 μl of anti-GFP-conjugated agarose beads (Chromotek), incubated at 4 °C for 3 h, and washed four times with washing buffer (20 mM HEPES, pH 7.4, 150 mM NaCl, 1 mM EDTA, 0.5% [v/v] Triton X-100, 1 mM PMSF, and 1×protease inhibitor cocktail). The bound proteins were eluted from the beads in 4× SDS-PAGE sample buffer and analyzed by immunoblot using anti-Myc (Proteintech, 60003-2-Ig, 1:5000(v/v)) or anti-GFP (Proteintech, 66002-1-Ig, 1:5000(v/v)) antibodies. The primers used in the generation of the relevant construct are listed in Supplementary Data 2.

## In vitro ubiquitination assay

The assay was performed as described previously with minor modifications[78]. Purified recombinant GmGA2ox-like-MBP (maltose binding protein) was used as substrate and GmILPA1-GFP as E3, GmILPA1 was immunopurified as the E3 from transgenic plants of 35S:GmILPA1-GFP using GFP-Trap agarose beads (Chromotek). The other ingredients, including E1, E2 (UbcH5b/UBE2D2), and ubiquitin (Ub), were purchased from Boston Biochem (Cambridge, MA). Ubiquitination assays were performed in reaction buffer containing 50 mM Tris-HCl, pH 7.5, 5 mM MgCl₂, 2 mM ATP, 2 mM dithiothreitol (DTT), and 20 μM MG132. After 2 h of incubation at 30 °C, the reaction was stopped by adding SDS loading buffer. Ubiquitinated GmGA2ox-like-MBP was detected by immunoblotting with anti-Ub (Proteintech, 10201-2-AP, 1:1000(v/v)) and anti-MBP (Proteintech, 66003-1-Ig, 1:5000(v/v)) antibodies. Determinant amino acids of GmGA2ox-like responsible for the ubiquitination were identified using an online prediction website (gpsuber.biocuckoo.cn/).

## In vivo ubiquitination assay

The assay was performed as described previously with minor modifications[79]. For the in vivo ubiquitination assay of GmGA2ox-like and GmILPA1, cultures of Agrobacterium strain GV3101 harboring the constructs *GmGA2ox-like-GFP* and *Ub-flag*, *GmGA2ox-like* and *GmUBL1*, *GmILPA1* and *GmGA2ox-like-GFP*, *GmGA2ox-like-GFP*, *GmGA2ox-like^K407R^-GFP*, *GmGA2ox-like^K394R^-GFP*, and *GmGA2ox-like^K394R,K407R^-GFP* were infiltrated into *N. benthamiana* leaves. 100 μM MG132 was co-injected with the *Agrobacterium* suspension to prevent the degradation of GmGA2ox-like-GFP protein. Leaf tissues were harvested 2 days after infiltration, and total proteins were extracted with extraction buffer (20 mM HEPES, pH 7.4, 150 mM NaCl, 1 mM EDTA, 0.5% [v/v] Triton X-100, 1 mM PMSF, 10% [v/v] glycerol, 1× protease inhibitor cocktail, and 0.1% [v/v] β-mercaptoethanol). The supernatant was mixed with 15 μl of anti-GFP-conjugated agarose beads (Chromotek) for 3 h at 4 °C. The beads were washed at least three times with washing buffer, followed by elution in 4× loading buffer and separation by SDS-PAGE and detection by immunoblotting using anti-FLAG (Proteintech, 66008-4-Ig, 1:1000(v/v)) or anti-ubiquitin antibodies (Proteintech, 10201-2-AP, 1:1000(v/v)). The primers used in the generation of the relevant constructs are listed in Supplementary Data 2

## Cell-free degradation assay

The assay was performed as described previously with minor modifications[80]. For the degradation of GmGA2ox-like in soybean, total proteins were extracted from Hedou 12 and *Gmilpa1-2* leaves using extraction buffer (50 mM Tris-MES, pH 8.0, 0.5 mM sucrose, 1 mM MgCl₂, 10 mM EDTA, and 5 mM DTT with freshly added protease inhibitor cocktail). After centrifugation for 15 min at 14,000 × $g$ at 4 °C. The protein extracts were then incubated with 10 mM ATP at 25 °C for 0 to 90 min in the presence or absence of 50 μM MG132. The abundance of GmGA2ox-like was determined by immunoblotting. For the degradation of GmGA2ox-like in *N. benthamiana*, total proteins extracted from leaves infiltrated with *GmILPA1-flag* or *GmUBL1-GFP* or from the leaves of non-infiltrated (control) *N. benthamiana* leaves were mixed with total proteins extracted from *N. benthamiana* leaves expressing *GmGA2ox-like-myc*. A final concentration of 10 mM ATP was added to the cell lysates to preserve the function of the 26 S proteasome before incubation for the indicated times at 25 °C. GmGA2ox-like-myc, GmILPA1-flag, and GmUBL1-GFP protein levels were analyzed by immunoblotting with anti-GFP (Proteintech, 66002-1-Ig, 1:5000(v/v)), anti-Myc (Proteintech, 60003-2-Ig, 1:5000(v/v)), and anti-Flag (Proteintech, 66008-4-Ig, 1:1000(v/v)) antibodies, respectively. The primers used in the generation of the relevant construct are listed in Supplementary Data 2.

## Affinity purification of ubiquitinated proteins

Total proteins of Wild-type and three P35S:GmGA2ox-like-GFP transgenic plants were extracted with 1 ml of BI buffer (50 mM Tris-HCl, (pH 7.5), 20 mM NaCl, 0.1% NP-40 and 5 mM ATP) in a prechilled mortar. The following were added to the protein homogenates: 1 mM PMSF, 50 mM MG132, 10 nM Ub aldehyde, and 10mM N-ethylmaleimide. Protein extracts were incubated with 40 μL of prewashed p62-agarose (Enzo Life Sciences, cat. no. BML-UW9010-0500) in 2 ml of BI buffer at 4 °C. After 4 h, the agaroses were washed two times with BI buffer and once with BII buffer (supplemented with 200 mM NaCl in BI). Samples were boiled in 50 μL of 1×SDS loading buffer for 5 min. The ubiquitinated proteins were separated by 10% SDS–PAGE gel, and anti-GFP antibody was used to detect ubiquitinated GmGA2ox-like-GFP protein.

## Haplotype of *GmILPA1*

For the investigation of the natural variation at *GmILPA1*, a panel of 444 soybean accessions consisting of 87 wild, 158 landrace, and 199 cultivated soybeans were subjected to haplotype analysis. A total of 10 SNPs and 1 InDel in the 5.7-kb region surrounding *GmILPA1* were used for haplotype classification. Six main haplotypes were obtained based on qualified genotypic and phenotypic data for *GmILPA1*. The remaining haplotypes were represented by fewer accessions, were deemed rare, and thus were not considered in this study. Haplotype network analysis was performed using PopART software based on the 'Median-Joining Network' approach. Phylogenetic analysis was based on UPGMA tree implemented in MEGA7 software.

## Detection of selection signals, nucleotide diversity, and tests for neutrality

The fixation index (FST), nucleotide diversity, and Tajima's D values in wild, landrace, and cultivated soybean germplasms were calculated across the ~3.7-kb genomic region of *GmILPA1* that includes the ~2.0-kb promoter, exons, introns, and 5' and 3' UTR sequences under a 100-kb sliding window analysis with a 5-kb step using VCFtools (v0.1.14).

## β-Glucuronidase (GUS) assay

The *GmILPA1pro^Hap1^:GUS*, *GmILPA1pro^Hap3^:GUS*, *GmILPA1pro^Hap5^:GUS*, *GmILPA1pro^Hap5/indel^:GUS* and *GmILPA1pro^Hap6^:GUS* reporter constructs were independently transformed into Agrobacterium strain GV3101 for transient expression via *Agrobacterium*-mediated infiltration. The plants were incubated for 3 days at 25 °C.

For UV-B treatment, the plants were treated with UV-B for 12 h before the infiltrated leaves were collected. *N. benthamiana* leaf discs were stained with 5-bromo-4-chloro-3-indolyl β-D-glucuronide (X-Gluc) for 24 h at 37 °C. For measurements of GUS activity, 4-methylumbelliferyl β-D-glucuronide (4-MUG) was added as substrate for a fluorometric assay, and the amount of 4-MU produced in the GUS reaction was measured on a Thermo Scientific Microplate Reader (Thermo Scientific). The total concentration of protein extracted from the leaf discs was measured using a BCA Protein Assay (Thermo Scientific) and diluted to a reaction concentration of 50 μg ml$^{-1}$. Final GUS activity was calculated according to a standard curve of 4-MU and expressed as pmol 4-MU/μg /min. The GUS assays were performed three times.

### Enzyme activity of recombinant GmGA2ox-like protein

The full-length cDNAs of GmGA2ox-like were cloned into the pEGX4T-2 vector. Construct was confirmed and transformed into *Escherichia coli* Transetta (DE3) (Transgene) for recombinant protein expression. The pEGX4T-2 vector was used as a control. For the enzyme activity assay, the procedure was adapted from previous studies with slight modification[81,82]. One hundred micrograms of total protein was incubated with GA metabolites (1 μg) in 100 μl of reaction mixture containing 100 mM Tris−HCl (pH 7.5), 1 mM FeSO4, 10 mM 2-oxoglutarate, 10 mM ascorbate, and 5 mM DTT at 30 °C for up to 6 h. After incubation, 100 μl of methanol was added and mixed. After centrifugation the supernatant was collected for analysis by LC−MS/MS (AB SECIEX API5500). A C18 HPLC capillary column (Agilent, 100 mm × 2.1 mm × 2.7 μm) was used for separation. Mobile phase was composed of solvent A (0.1% formic acid) and solvent B (100% MeOH). Solvent gradient was programmed as 15% of solvent B for 2 min, increasing solvent B to 95% over 15 min, holding for 3 min, and returning back to 5% of solvent B over 2 min, followed by a 5-min re-equilibration prior to the next sample injection. Flow rate was maintained at 0.3 ml/min. Each sample injection volume was 50 μl. MS spectra were acquired in negative electrospray ionization combined with the SRM (selected-reaction monitoring) mode and analyzed using software (Analyst).

### Statistics and reproducibility

Significant differences were analysed using two-sample Student's *t*-tests or one-way ANOVA with GraphPad Prism 8.0. All experiments were repeated independently three times.

### Reporting summary

Further information on research design is available in the Nature Portfolio Reporting Summary linked to this article.

## Data availability

Data supporting the findings of this work are available within the paper and its Supplementary Information files. The SNP/Indel data of natural population accessions were previously reported were deposited into the NCBI database under accession number PRJNA394629 and the Genome Sequence Archive database in BIG Data Center under accession numbers PRJCA000205 and PRJCA001691. The Wm82 a4.v1 reference genome was download from Phytozome. Source data are provided with this paper.

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

## Acknowledgements

The authors thank Haiyan Yu, Xiaomin Zhao and Sen Wang from SKLMT (State Key Laboratory of Microbial Technology, Shandong University) for the assistance in microimaging of LSCM analysis. This research was supported by the National Key Research and Development Program of China (grant 2021YFF1001201), the Joint Funds of the National Natural Science Foundation of China (grant U1906203), National Key Research and Development Program of China (2022YFF1001601-4), Key Research and Development Program of Shandong Province (grant 2021LZGC003), the National Transgenic Project of China (grants 2018ZX08009-14B and 2016ZX08010002-009), Taishan Scholar Youth Project of Shandong Province and Qingdao Natural Science Foundation (23-2-1-38-zyyd-jch) for Q.L. Thanks to Plant Editor for language editing.

## Author contributions

F.X. conceived the project. J.S., F.X., Q.L. and S.L. designed the experiments. J.S. and S.H. drafted the manuscript. J.S. performed most of the experiments. S.H. performed haplotypes analyses. S.Z. and X.Z. provided the inbred population. S.Z., Q.Z., W.Zha., J.L., L.W., W.Zhe. and K.Z. performed parts of the biochemical, phenotyping, and transformation experiments. F.X. and Q.L. revised the manuscript. X.F. provided mutant allele and some suggestions. F.K. and B.L. provided sequencing data of soybean germplasm resources and some suggestions. All authors read and approved of its content.

## Competing interests

The authors declare no competing interests.
