## [Peer Review File · Nature Communications]

UV-B irradiation-activated E3 ligase GmILPA1 modulates gibberellin catabolism to increase plant height in soybeanREVIEWER COMMENTS

Reviewer #1 (Remarks to the Author):

The manuscript describes the identification of a component of a ubiquitin E3 ligase from a forward genetics mutant screen for factors involvement in height determination in soybean. A mutagenized dwarfed plant was obtained from the screen and the associated gene GmUID1, identified by mapping, encodes a subunit of an APC/C-type ubiquitin ligase. A yeast 2-hybrid screen identified a gibberellin 2-oxidase-like (GA2ox) protein as a substrate for GmUID1 and evidence is presented that it is ubiquitinated by the ligase and targeted for degradation by the 26S proteasome. This reveals a novel, post-translational mechanism for the regulation of gibberellin metabolism. The dwarf phenotype occurs only in UV-B light, which stimulates the interaction of the GA2ox with GmUID1 by an unknown mechanism. Furthermore, expression of GmUID1 is promoted by UV-B. A second potential E3 ligase was identified from the yeast 2-hybrid screen and shown to interact with both GmUID1 and the GA2ox, enhancing ubiquitination of the GA2ox. Interestingly, evidence is presented for selection during soybean domestication for a GmUID1 haplotype lacking a light-responsive element in the promoter so reducing its response to UV-B-stimulated expression and promoting reduced stature. The work contains considerable potential novelty, but more evidence is required to support the conclusions.

The effect of the mutation on stature is surprisingly strong, more than I would expect from the presence or absence of a GA2ox, as there are numerous GA2ox paralogues with considerable redundancy. Can the authors be sure that the dwarf phenotype is due solely to stabilising this GA2ox rather than other GmUID1 substrates?

It is important to characterise the GA2ox and confirm its function. It is shown to be a homolog of the Arabidopsis GA2ox7, i.e. a C20-GA2ox. This should be better documented by showing a phylogenetic tree which includes the other AtGA2ox genes. Ideally its biochemical function would be demonstrated, for example using recombinant enzyme in vitro.

If it is a C20-GA2ox rather than a C19-GA2ox, then it would act early in the GA-biosynthetic pathway and GA51 levels should be lower rather than higher as shown for the Gmuid1 mutant. GA3 is not a major plant GA and is not metabolised by GA2ox. The authors need to measure other more relevant GAs, such as GA1 and GA4, as well as precursors and catabolites.

The method description for the GA analysis is inadequate and more information is required. It describes using GC-MS, but no conditions are provided. It refers to a paper in which LC-MS was used which is not helpful.

Other points:

Lines 47-51: Yields were increased by using high-yielding varieties and applying high levels of fertiliser in combination with the introduction of semi-dwarfing genes to prevent lodging. The genes also improved harvest index.

Line 58: gibberellin rather than gibberellic acid, which is specifically GA3.

Lines 64-65: what is meant by central hub modulators?

Lines 163-166: the mutation modifies a splice site with the assumption that the intron is transcribed with the introduction of a stop codon. Was this confirmed by sequencing the mutant transcripts?

Line 326: It is not simply the balance between GA3 and GA51. Other more relevant GAs need to be analysed.

Line 387: The HAP5 phenotype with reduced nodes and branches is not typical of lower gibberellin levels.

Reviewer #2 (Remarks to the Author):

Sun et al. report the soybean mutant Gmuid1 as being dwarf specifically in the field or in the greenhouse under UV-B. GmUID (previously identified and named by Gao et al., 2017 as GmILPA1) is

an APC8-like protein, a subunit of the E3 ubiquitin ligase APC/C. GmUID1 apparently ubiquitinates the GA-inactivating enzyme GmGA2ox-like under UV-B, resulting in the degradation of GmGA2ox-like. This is further enhanced by another E3 ubiquitin ligase identified by the authors, namely GmUBL1. In agreement with the finding that GmUID1 ubiquitinates GmGA2ox-like, the Gmuid1 mutant shows reduced levels of active GA3 (as the catabolic GmGA2ox-like levels would be enhanced in absence of its E3 ubiquitin ligase, resulting in enhanced inactivation of active GA, and thus dwarf growth). What is not clear, however, is how these findings fit to UV-B regulation of active GA levels and growth in wild type. The work by the authors suggests that UV-B (likely by UVR8 photoreceptor signaling) should lead to elevated levels of active GA (as GmUID1 is activated, thus GmGA2ox-like levels reduced – i.e. less active GA being catabolized) and thus there should be enhanced growth under UV-B. However, available data, mainly from Arabidopsis, but also other plants, point to UVR8-dependent growth inhibition under UV-B, associated with reduced levels of GA and elevated DELLA levels. The authors should address this discrepancy. At present no data is provided how wild-type soybean responds to UV-B in the field, in regard of growth as well as GA levels. The manuscript provides potentially very interesting findings, but the below points need to be addressed (particularly also point 1) before publication.

Major points:

1) UV filters should be used in the field to support and extend the findings, comparing wild type under UV and no-UV conditions (growth and GA levels under UV exclusion versus under a UV transmitting mock filter). Same for Gmuid1 mutant (i.e. rescue of phenotypes by filtering out UV in the field which would provide very strongly support that indeed UV is the underlying reason for the Gmuid1 phenotype in the field). Is UV-B indeed resulting in higher active GA levels and enhanced growth in soybean, as suggested but not investigated nor discussed in this work? If so this would be in contrast to findings in other plants, and possibly even soybean (see e.g. also Vanhaelewyn et al J.Exp.Bot. 2016, Hayes et al PNAS 2014), and thus should be well documented. Also well-controlled UV-B experiments in the glasshouse can be additionally performed.

2) The finding that UV-B promotes the interaction between GmUID1 and GmGA2ox-like should be supported by Co-IP experiments, and not be limited to BiFC experiments (Fig. 3h,i). This should be doable as Co-IPs in absence of UV-B are provided in Fig. 3b,d,f. GmUID1 – GmUBL1 and GmUBL1 – GmGA2ox-like provide control Co-IPs that should not be affected by UV-B.

3) Clarify that GmUID1 is GmILPA1, and that the Gmuid1-2 has been published before as Gmilpa1 mutant.

Line 17: "...H12 named as Gmuid1, the other name Gmilpa1...". Not clear... Shouldn't the name previously given be kept? That would be clearer than having several names for the same gene. Is Gmuid1-2 the previously published Gmilpa1 mutant (Gao et al., Plant Physiol 2017)? Seems like from the description of the mutation (1149-bp deletion incl. 4th exon...). Thus why "... we isolated additional mutant allele...". Isolated from where if already published. Please clarify.

Lines 181-182: "Sequence comparison and phylogenetic analysis demonstrated that GmUID1 encodes an APC8-like protein, ...", rather confirmed, if at all, as the gene has already been published before (Gao et al 2017 entitled "GmILPA1, Encoding an APC8-like Protein, Controls Leaf Petiole Angle in Soybean")

Line 187: "..., which revealed that GmUID1-GFP localizes to both the nucleus and cytoplasm..."; also confirmed, as already shown by Gao et al., 2017 for GmILPA1, which = GmUID1...

4) Please add western analysis of GmGA2ox-like protein levels in wild-type under UV and no-UV conditions, as well as same for Gmuid1. An anti-GmGA2ox-like antibody is available, but was only used in cell-free degradation assays (Fig. 4i,j). An effect on GmGA2ox-like levels should be tested in western analyses as well. Elevated levels in Gmuid1? Reduced levels in wild type under UV versus no-UV?

5) Fig 5c: discuss why +GA only suppresses dwarfism of Gmuid1, but results in strongly changed plant morphology compared to wild type.

Minor points:

- Line 24: "gibberellic acid (GA)"; consistent line 58.
- Line 36: What is meant by "..., effectively improving GmUID1 function."; not clear, please rephrase.
- lines 924-1101: Several references listed twice; please recheck.
- Line 77: "UV RESISTANCE LOCUS8 (UVR8)"
- Line 110: "The other E3 ubiquitin ligase, ..." suggests there are only two E3 ligases in soybean. Please rephrase.
- Line 114: "affection"? Entire sentence not clear to me...
- Line 119/120: "... might be due to the damage of cis-regulatory elements". Rather: due to specific mutations in cis-regulatory elements?
- Line 141: "..., 153 showed the wild-type phenotype"; add "wild type"?
- Line 165: "new increased exon"? "exon of the protein"? Please correct.
- Fig. 4c: why the smear also below the band if indicating ubiquitination of GmGA2ox-like-MBP?
- Line 315: should likely read "... under white light in the field (Fig. 5c-d)". In general, please describe conditions clearer in figure legends.
- Line 321: in contrast to statement "... within 12 h of combined treatment with GA3 and UV-B light (Fig. 5g-h)" apparently no combined treatment shown in Fig. 5g-h.
- Fig. 5j: n=? Field experiment?
- Lines 454-454: "... relative GmUID1 transcript activity were higher in accessions harboring..."? This statement seems to refer to GUS assays in *N.benthamiana*... Needs to be corrected.
- Line 472: Light and growth conditions? How old plants? How long exposed to UV-B?
- Fig. 2 b-h. Conditions are not clearly described, but it seems 2b very different to 2d-h. Re-arrange figure that one does not compare plant height in 2b versus others measurements. As they are directly next to each other, it is rather confusing. Mention the light treatments in the legend (duration, fluence rate).
- Fig. 4a: why is there such a strong difference in the input between - and + GmGA2ox-like-GFP on total Ub-Flag?
- Lines 739 – 740: provide fluence rates in $\mu\text{mol m}^{-2} \text{sec}^{-1}$.

Reviewer #3 (Remarks to the Author):

Sun et. al. provides a new hypothesis on the post-translational regulation of GA2ox-like in a UVB-dependent manner in Soybean. This is an intriguing possibility which was not been explored before. The authors also tried to provide evidence for the E3 ligases that ubiquitinate the GA2ox-like. In addition, the prevalence of different haplotypes in one of the E3s, the GmUID1 has been investigated and proposed that a defect in its cis-element may have contributed to a UVB-dependent plant height phenotype.

Even though the manuscript provides a large amount of experimental data, I have some serious concerns over the ways in which the experiments were conducted and the rationale behind the explanation and interpretation of the results. I have the following major and minor comments that I request the authors to address in detail:

Major:

1. The authors provide Gmilpa1 as a mutant allelic to Gmuid1-1. Gmilpa1 has been published previously (ref. 53 in the manuscript). However, Gmilpa1 do not show a dwarf phenotype (Figure 1 and 3 of ref. 53). Can the authors please clarify this discrepancy?

2. Even though the authors show data for the mutual interaction between GmUID1, GmUBL1 and GmGA2-ox-like, it is not clear how these interactions ultimately cooperate in GmGA2ox-like degradation in presence of UVB. There are several questions remain unanswered:

- a. What is the role of GmUBL1 in GmUID1-GmGA2ox-like interaction influenced by UVB?
- b. What are the respective contributions of GmUBL1 and GmUID1 in ubiquitinating GmGA2ox-like?
- c. What is the relevance of direct interaction between GmUBL1 and GmUID1?

3. The experiments involving different wavelengths of light are not convincing

- a. UVB is often detrimental to plant growth, so weak narrow-band UVB coupled with white light is preferred, especially at the seedling stage. However, the authors have used intense UVB radiation alone (methods section; Lines 739-741). This could induce potential stress on the plants. Moreover, the exact experimental conditions used are unclear
- b. For Fig. 2C-2H: The quantity, duration and frequency of light treatment were not mentioned. It is also important to mention the ZT at which the light treatments were given, as this affects the photoreceptor activities
- c. The methods section (Lines 739-742) shows that the UVB treatment was only for 2 hours and the plants were returned to white light. Irradiation with the other wavelength is also apparently performed in a similar fashion. It is surprising that such a short treatment was able to produce the seedling phenotypes shown in Fig 2C. Again, the experimental conditions are not well explained
- d. It is not clear how the UVB experiments were performed for Figures 4i, 5e, 5h, 6h, Ext. data Fig. 7 and Ext. data Fig. 8

4. There are multiple problems with the protein interaction experiments

- a. Lane 4 and Lane 8 of the Y2H data in Fig. 3A: Seem to be duplicated
- b. The authors have used 3-AT presumably because the BD-GmUID1 showed autoactivation. A fold-dilution series should be performed in this scenario to show the interaction strength. Without this, the results can only be interpreted as a consequence of autoactivation
- c. Using the yeast system, the UVB-dependency of the interactions could easily be tested. I suggest the authors perform the Y2H interaction assays in a quantitative manner in the presence or absence of UVB
- d. Co-IP experiments were performed in Nicotiana, while the authors already have transgenic lines expressing GFP-tagged proteins. These lines could be used effectively coupled with agrobacterium-infiltration (King et. al., 2015) for all the Co-IP and protein degradation experiments shown in the manuscript
- e. Moreover, the authors have produced specific antibodies that detect the Gm proteins. Then why use tagged proteins in Nicotiana?
- f. Equal loading controls for the input samples are missing in the Co-IP experiments
- g. Co-IP experiments were shown in chopped images. Hence the original blots must be provided as additional data
- h. In general, proper controls are missing from BiFC experiments. Ideal experiments must include positive and negative controls that improve reliability (Horstman et. al., 2014)
- i. The strength of interaction of GmGA2ox-like and GmUID1 under white light in Fig. 3C is much stronger than the one shown in Fig. 3H. Can the authors comment on this?
- j. In Fig. 3H, the method of quantification, laser intensity and exposure time used, UVB treatment and duration should be provided
- k. Figure 3H-3I: It is not clear why the interaction has been observed only in the cytoplasm and not in the nucleus, since GmUID1 also localizes strongly in the nucleus (Ext. Fig. 5C). Also, the subcellular localization of GmUBL1 and GmGA2ox-like was also not shown. Subcellular localization of all three proteins would help in interpreting the data shown in Fig. 3i

5. Discrepancies in the ubiquitination assay. The experiments are not convincing enough for establishing GmGA2ox-like as a substrate of GmUID1

- a. The authors resort to using Nicotiana, while they have created Gm transgenic plants. Please see

also comment 5a

b. Fig. 4a: The misexpressed Gm proteins are shown to be ubiquitinated in Nicotiana. Are there any orthologues in Nicotiana that act as E3? If so, is it also UVB-dependent? Ideally, GmUID1 should also be co-infiltrated to test ubiquitination. Alternatively, the GFP-tagged Gm line should be used for ubiquitination assay

c. Authors have used an anti-Ub antibody in Fig. 4B but infiltrated Ub-FLAG in Figure 4A. Why not use anti-Ub in both experiments?

d. Fig. 4A: Blot with anti-GFP should also be shown. Loading controls are missing for the input samples. As ubiquitinated proteins are degraded while doing the experiments, MG132 is often used. It is not clear how this experiment was performed in Nicotiana

e. Fig. 4B: The input samples in lanes 3 and 4 seem to be low and that may be the reason for the lower ubiquitination.

f. Fig. 4C: GmUID1 seems to be auto-ubiquitinated. However, this is not addressed anywhere in the manuscript. To have more reliability, the K394R version should be included in the in vitro ubiquitination assay

g. Fig. 4D: There is no explanation of the ways in which the experiment was conducted. It is not clear why the time points were shown or what to expect from this figure

h. It is surprising that the GmGA2ox-like-GFP infiltrated in Nicotiana shows strong ubiquitination in Figure 4A, while it shows weak ubiquitination in Figure 4e. How would the authors explain this?

i. Fig. 4F and 4G: The details of UVB experiments are missing

j. Fig. 4i and 4J: Complete blots should be provided as additional data

k. In Fig. 4J, blot at the end of the right-hand side, last lane: This is important data that shows the degradation of GmGA2-ox-like. However, the Ponceau staining shows that fewer proteins were loaded in this well

l. Fig. 4i and 4J: Since the protein degradation is apparent from 90 minutes, the time points for the experiment should be extended up to 150 minutes, which will show the degradation dynamics. This is especially relevant since the UVB-mediated enhancement in GmUID1 could be observed only after at least 3-4 hours of UVB exposure (Fig. 5G)

m. Since GmUID1 targets the D-BOX in GmGA2ox-like, a D-BOX mutated version should be used for the ubiquitination assays in order to verify that GmUID1 acts as an E3 in degrading GmGA2ox-like

6. For the UVB-induced phenotype in Fig. 2C-2H, only Gmuid1-1 is used. Is there any specific reason? Since the UVB phenotype is the basis for further experiments in the manuscript, the UVB phenotype of Gmuid1-2/Gmilpa1 and also the F1 plants resulting from the cross Gmuid1-1 x Gmuid1-2 should be used. This is required to establish the role of GmUID1 as an enhancer of plant height under UVB

7. To complement the Gmuid1-1 mutant, the authors resorted to overexpressing the full-length cDNA tagged with GFP. Ideally, the native promoter driving full-length genomic DNA should be used for complementation

8. Specific antibodies used in experiments such as the ones shown in Fig. 4 need validation experiments. The authors could easily show this via western blots using the WT, overexpression lines and mutants. What protein sizes are detected by these antibodies? Are there any non-specific bands detected? Please provide this as additional data

9. In Fig. 5A and 5B: GA51 content was not registered in the WT. Normally a basal level of GA degradation is expected in WT. Can this be explained?

10. Fig. 5E: Please clarify why PAC along with GA was used in this UVB experiment

11. Fig. 6H: The difference in GUS activity seems to be rather due to different infiltration efficiency. There are no infiltration controls provided in this experiment

12. For the GUS assay of haplotypes, an ideal control could be hap5 with the INDEL. Moreover, it

would be interesting to see the phenotype of hap5 in the presence or absence of UVB

13. Figure 6J: The plant height phenotype of haplotypes should be provided

14. Ext. data fig. 1C and 1D: The number of replicates and the number of cells per replicate used for the experiment should be stated. Microscopy procedures should be added to the methods section

Minor:

1. Lines 26-27: APC/C is abbreviated in the abstract without providing the full form
2. The effect of GA on UVB-induced hypocotyl elongation has recently been shown in Arabidopsis (Miao et. al., 2021). The results should be discussed in this context
3. Line 114: Do the authors mean the effect of UVB and GA?
4. Mostly the form "UV-B" is used in the manuscript. However, in some places "UVB" is used. I suggest making the word uniform throughout
5. Include microscopy procedures followed in the methods section
6. Lines 140-141: Do the authors mean 150 plants with WT phenotype?
7. Lines 154-155: The sentence lack information on NGS used in the bulk segregant analysis
8. In Fig. 3, C, E and G could be combined with common controls
9. Line 252: The bioinformatic tool used should be specified in the methods section
10. Line 295: Space between proteasome and in

Reviewer #1 (Comments for the Author):

The manuscript describes the identification of a component of a ubiquitin E3 ligase from a forward genetics mutant screen for factors involvement in height determination in soybean. A mutagenized dwarfed plant was obtained from the screen and the associated gene GmUID1, identified by mapping, encodes a subunit of an APC/C-type ubiquitin ligase. A yeast 2-hybrid screen identified a gibberellin 2-oxidase-like (GA2ox) protein as a substrate for GmUID1 and evidence is presented that it is ubiquitinated by the ligase and targeted for degradation by the 26S proteasome. This reveals a novel, post-translational mechanism for the regulation of gibberellin metabolism. The dwarf phenotype occurs only in UV-B light, which stimulates the interaction of the GA2ox with GmUID1 by an unknown mechanism. Furthermore, expression of GmUID1 is promoted by UV-B. A second potential E3 ligase was identified from the yeast 2-hybrid screen and shown to interact with both GmUID1 and the GA2ox, enhancing ubiquitination of the GA2ox. Interestingly, evidence is presented for selection during soybean domestication for a GmUID1 haplotype lacking a light-responsive element in the promoter so reducing its response to UV-B-stimulated expression and promoting reduced stature.

The work contains considerable potential novelty, but more evidence is required to support the conclusions.

1. The effect of the mutation on stature is surprisingly strong, more than I would expect from the presence or absence of a GA2ox, as there are numerous GA2ox paralogues with considerable redundancy. Can the authors be sure that the dwarf phenotype is due solely to stabilising this GA2ox rather than other GmUID1 substrates?

Response: Thank you for your suggestion. By examination of the phenotype of *Gmuid1-1/GmGA2ox-like RNAi* plants, our results strongly suggest that the essential role GmGA2ox-like in GmUID1-mediated plant height under UV-B irradiation. In order to estimate that the dwarf phenotype of *Gmuid1-1* observed is solely due to GA2ox stabilization, we obtained the transgenic RNA interference lines of *GmGA2ox-like* in *Gmuid1-1* (*Gmuid1-1/GmGA2ox-like RNAi*) in T₂ generation, and grew H12, *Gmuid1-1* and *Gmuid1-1/GmGA2ox-like RNAi* plants under white light

supplemented with UV-B. We found that the height of *Gmuid1-1/GmGA2ox-like RNAi* plants were notably higher than that of the *Gmuid1-1* (please see the new Fig. 5l), indicating that *GmGA2ox-like RNAi* rescued the dwarf phenotype of *Gmuid1-1*. Our results strongly suggest that GmGA2ox-like plays an essential role in GmUID1-mediated plant elongation under UV-B irradiation, although we still could not conclude that the dwarf phenotype of *Gmuid1* is solely due to GmGA2ox-like stabilization. These results have been introduced in the revised manuscript (please see the lines 364-370).

2. It is important to characterise the GA2ox and confirm its function. It is shown to be a homolog of the Arabidopsis GA2ox7, i.e. a C20-GA2ox. This should be better documented by showing a phylogenetic tree which includes the other AtGA2ox genes. Ideally its biochemical function would be demonstrated, for example using recombinant enzyme in vitro.

Response: Thank you for your valuable suggestion. We re-performed the phylogenetic tree with other AtGA2ox genes (please see the new Extended Data Fig. 6a), and found that GmGA2ox-like is close homolog of Arabidopsis GA2ox4, which belong to C19-GA2ox rather than a C20-GA2ox. This result has been added in the revised manuscript (please see the lines 208-209).

3. If it is a C20-GA2ox rather than a C19-GA2ox, then it would act early in the GA-biosynthetic pathway and GA51 levels should be lower rather than higher as shown for the *Gmuid1* mutant. GA3 is not a major plant GA and is not metabolised by GA2ox. The authors need to measure other more relevant GAs, such as GA1 and GA4, as well as precursors and catabolites.

Response: Thank you for your suggestion. We measured the contents of endogenous GAs (GA1, GA4, GA20, GA9, GA8 and GA34) in H12 and *Gmuid1-1* under white light alone and white light supplemented with UV-B. The results showed no difference in GAs content between H12 and *Gmuid1-1* under white light (please see the new Fig. 5e-j), while the gibberellin GA1, GA4, GA20 and GA9 were more abundant in the WT than in *Gmuid1-1* under white light supplemented with UV-B, the non-bioactive GA8 and GA34 accumulated at a relatively high level in *Gmuid1-1* relative to the H12 (please see the new Fig. 5e-j). These results also provide important evidences that GmGA2ox-like belong to a C19-GA2ox. All the results have been added in the revised manuscript (please see the lines 342-355).

4. The method description for the GA analysis is inadequate and more information is required. It describes using GC-MS, but no conditions are provided. It refers to a paper in which LC-MS was used which is not helpful.

Response: Thank you so much for pointing out this problem. The method description for the GA analysis has been revised as following: To quantify the contents of GAs, H12, *Gmuid1-1* seedlings were grown under white light and white light with UV-B. The stem apexes of the seedlings were collected, frozen in liquid nitrogen, and ground to a fine powder, and extracted with the traction method (methanol/water/formic acid = 15:4:1, V/V/V). The extracts were vortexed and centrifuged at $4694 \times g$ under 4°C for 10min. The supernatants were dried by evaporation under the flow of nitrogen gas at room temperature, then dissolved in 200 μl of methanol. The sample extracts were analyzed using an LC-ESI-MS/MS system (HPLC, Shimpack UFLC SHIMADZU CBM30A system;MS, Applied Biosystems 6500 Triple), and the data were analyzed by Metware Biotechnology Co., Ltd. Wuhan, China. At least three replicates of each assay were performed (please see the lines 902-912).

5. Other points:

Lines 47-51: Yields were increased by using high-yielding varieties and applying high levels of fertiliser in combination with the introduction of semi-dwarfing genes to prevent lodging. The genes also improved harvest index.

Response: It has been revised as you suggested. We added the description about the harvest index, we have revised the sentences (please see the line 51).

Line 58: gibberellin rather than gibberellic acid, which is specifically GA3.

Response: Thank you for pointing out this problem. We have exchanged gibberellic acid to gibberellin in the revised manuscript (please see the line 58).

Lines 64-65: what is meant by central hub modulators?

Response: To make the sentence more concise, we have changed the description of “central hub modulators” to “Loss-of-function mutations in GA biosynthetic genes such as *GA20ox* and *GA3ox*

often lead to dwarfism” (please see the lines 63-64).

Lines 163-166: the mutation modifies a splice site with the assumption that the intron is transcribed with the introduction of a stop codon. Was this confirmed by sequencing the mutant transcripts?

Response: Thank you for your suggestion. We added the sequencing results for *Glyma.11G026400* in wild type and *Gmuid1-1* mutant. Please see the new Extended Data Fig. 3a.

Line 326: It is not simply the balance between GA3 and GA51. Other more relevant GAs need to be analyzed.

Response: We do agree that more GAs should be analyzed. Thus, we measured the contents of endogenous GAs (GA1, GA4, GA20, GA9, GA8 and GA34) in H12 and *Gmuid1-1* under white light and white light supplemented with UV-B conditions. We detail the results in our response to the Editorial comments 2 and the Result section in the revised manuscript (please see the lines 342-355).

Line 387: The HAP5 phenotype with reduced nodes and branches is not typical of lower gibberellin levels.

Response: Thank you for your suggestion. To focus on the regulation of gibberellin, we deleted the phenotypes with node numbers and branches in different haplotypes (please see the new fig. 6j).

Reviewer #2 (Remarks to the Author):

Sun et al. report the soybean mutant *Gmuid1* as being dwarf specifically in the field or in the greenhouse under UV-B. *GmUID1* (previously identified and named by Gao et al., 2017 as *GmILPA1*) is an APC8-like protein, a subunit of the E3 ubiquitin ligase APC/C. *GmUID1* apparently ubiquitinates the GA-inactivating enzyme *GmGA2ox*-like under UV-B, resulting in the degradation of *GmGA2ox*-like. This is further enhanced by another E3 ubiquitin ligase identified by the authors, namely *GmUBL1*. In agreement with the finding that *GmUID1* ubiquitinates *GmGA2ox*-like, the *Gmuid1* mutant shows reduced levels of active GA3 (as the catabolic *GmGA2ox*-like levels would be enhanced in absence of its E3 ubiquitin ligase, resulting in enhanced inactivation of active GA, and thus dwarf growth). What is not clear, however, is how these findings fit to UV-B regulation of active GA levels and growth in wild type. The work by the authors suggests

that UV-B (likely by UVR8 photoreceptor signaling) should lead to elevated levels of active GA (as GmUID1 is activated, thus GmGA2ox-like levels reduced – i.e. less active GA being catabolized) and thus there should be enhanced growth under UV-B. However, available data, mainly from Arabidopsis, but also other plants, point to UVR8-dependent growth inhibition under UV-B, associated with reduced levels of GA and elevated DELLA levels. The authors should address this discrepancy. At present no data is provided how wild-type soybean responds to UV-B in the field, in regard of growth as well as GA levels. The manuscript provides potentially very interesting findings, but the below points need to be addressed (particularly also point 1) before publication.

Response: Thank you for your critical comments, which have been very helpful in improving the quality of the manuscript.

Major points:

1. UV filters should be used in the field to support and extend the findings, comparing wild type under UV and no-UV conditions (growth and GA levels under UV exclusion versus under a UV transmitting mock filter). Same for Gmuid1 mutant (i.e. rescue of phenotypes by filtering out UV in the field which would provide very strongly support that indeed UV is the underlying reason for the Gmuid1 phenotype in the field). Is UV-B indeed resulting in higher active GA levels and enhanced growth in soybean, as suggested but not investigated nor discussed in this work? If so this would be in contrast to findings in other plants, and possibly even soybean (see e.g. also Vanhaelewyn et al J.Exp.Bot. 2016, Hayes et al PNAS 2014), and thus should be well documented. Also well-controlled UV-B experiments in the glasshouse can be additionally performed.

Response: Thank you for your suggestion. We do agree that filtering out UV in the field could strongly support and extend our findings. However, it is not suitable for field cultivation currently. To deal with this problem, we performed well-controlled UV-B experiments in the glasshouse using narrow-band UV-B as suggested here and by Reviewer 3, which was commonly used in the UV-B signaling field (Podolec, R. et al PNAS 2020; Podolec, R. et al The plant journal 2022; Yu Yang, et al Nature plants 2018). In detail, H12 and *Gmuid1-1* plants were grown under white light (600 $\mu\text{mol m}^{-2} \text{s}^{-1}$, 12 h /12 h light/dark cycles) or white light supplemented with UV-B (1.5 $\mu\text{mol m}^{-2} \text{s}^{-1}$, 12 h /12 h light/dark cycles, TL20W/01RS tubes; Philips).

To further clarify the role of GmUID1-GmGA2ox-like-mediated GA catabolism in plant growth

under UV-B irradiation, first we measured the contents of endogenous GAs (GA₁, GA₄, GA₂₀, GA₉, GA₈ and GA₃₄) in H12 and *Gmuid1-1* plants under white light alone and white light supplemented with UV-B. In general, the endogenous GA content in both H12 and *Gmuid1-1* plants showed a decreasing trend under white light supplemented with UV-B (please see the new Fig. 5e-j), which demonstrated that UV-B reduced the GA level in soybean and is consistent with the previous studies (Vanhaelewyn et al J.Exp.Bot. 2016, Hayes et al PNAS 2014). However, the reduction rate of active GA in *Gmuid1-1* was higher than that in H12, indicating that GmUID1 is important in UV-B-regulated active GA levels. In addition, in consistent with the phenotype observed, there was no difference in the levels of GAs between H12 and *Gmuid1-1* under white light, while the gibberellin GA₁, GA₄, GA₂₀ and GA₉ were more abundant in the WT plants than in *Gmuid1-1* under white light supplemented with UV-B conditions, the non-bioactive GA₈ and GA₃₄ accumulated at a relatively high level in *Gmuid1-1* relative to the H12 under same conditions (please see the new Fig. 5e-j). Therefore, our results demonstrated that GmUID1 repressed the UV-B-induced decreasing of active GA levels to improve the UV-B tolerance in soybean. Consistently, UV-B irradiation reduced the protein levels of GmGA2ox-like in WT but not in *Gmuid1-1* (please see the new Fig. 4g). Together, our results suggested that GmUID1-GmGA2ox-like module enhances the UV-B tolerance in soybean by repressing UV-B-induced reduction of active GA levels.

2. The finding that UV-B promotes the interaction between GmUID1 and GmGA2ox-like should be supported by Co-IP experiments, and not be limited to BiFC experiments (Fig. 3h,i). This should be doable as Co-IPs in absence of UV-B are provided in Fig. 3b,d,f. GmUID1 – GmUBL1 and GmUBL1 – GmGA2ox-like provide control Co-IPs that should not be affected by UV-B.

Response: We agree that Co-IP should also be done to support our conclusion. Therefore, we performed the Co-IP experiments in presence or absence of UV-B. In consistent with BiFC experiments, Our Co-IP results showed that UV-B promoted the interaction between GmUID1 and GmGA2ox-like, but did not promote the interaction between GmUID1 and GmUBL1, GmUBL1 and GmGA2ox-like (please see the new Fig. 3h, Extended Data Fig. 7f, 7i).

3. Clarify that GmUID1 is GmILPA1, and that the *Gmuid1-2* has been published before as *Gmilpal* mutant.

Line 17: "...H12 named as *Gmuid1*, the other name *Gmilpa1*...". Not clear... Shouldn't the name previously given be kept? That would be clearer than having several names for the same gene. Is *Gmuid1-2* the previously published *Gmilpa1* mutant (Gao et al., Plant Physiol 2017)? Seems like from the description of the mutation (1149-bp deletion incl. 4th exon...). Thus why "... we isolated additional mutant allele...". Isolated from where if already published. Please clarify. Lines 181-182: "Sequence comparison and phylogenetic analysis demonstrated that GmUID1 encodes an AP8C-like protein, ...", rather confirmed, if at all, as the gene has already been published before (Gao et al 2017 entitled "GmILPA1, Encoding an APC8-like Protein, Controls Leaf Petiole Angle in Soybean")

Line 187: "..., which revealed that GmUID1-GFP localizes to both the nucleus and cytoplasm..."; also confirmed, as already shown by Gao et al., 2017 for GmILPA1, which = GmUID1...

Response: Thank you for your suggestion. We thought that *GmUID1* is a UV-B-induced dwarf-related gene, so, GmUID1 named is probably more appropriate for the main idea of this manuscript as title "UV-B irradiation-activated E3 ligase GmUID1 modulates gibberellin catabolism to increase plant height in soybean". Therefore, we didn't exchange *GmUID1* to *GmILPA1* in new revision. If you don't agree with us, we'll exchange *GmUID1* to *GmILPA1*.

4. Please add western analysis of GmGA2ox-like protein levels in wild-type under UV and no-UV conditions, as well as same for *Gmuid1*. An anti-GmGA2ox-like antibody is available, but was only used in cell-free degradation assays (Fig. 4i,j). An effect on GmGA2ox-like levels should be tested in western analyses as well. Elevated levels in *Gmuid1*? Reduced levels in wild type under UV versus no-UV?

Response: Thank you for your suggestion. We analyzed the GmGA2ox-like protein levels in H12 and *Gmuid1-1* under UV-B and no-UV-B conditions. The results showed that the protein levels of GmGA2ox-like were similar between H12 and *Gmuid1-1* under white light. However, UV-B irradiation reduced the amount of GmGA2ox-like in H12 but not in *Gmuid1-1* plants, demonstrating that UV-B irradiation regulated GmGA2ox-like stabilization via GmUID1 (please see the new Fig. 4g).

5) Fig 5c: discuss why +GA only suppresses dwarfism of *Gmuid1*, but results in strongly changed

plant morphology compared to wild type.

Response: Thank you for your suggestion. Our findings showed that GmUID1 regulated plant height under UV-B irradiation. However, we speculate that GmUID1 might be a multiple-function E3 ligase that involved in different growth and development processes via different mechanisms. For example, the Leaf petiole Angle in *Gmuid1* is larger than that of H12 (Extended Data Fig. 1a, Gao et al Plant Physiol 2017), indicating that *GmUID1* also controls leaf petiole angle in soybean. Therefore, treatment with GA3, only restored plant height, but not other phenotype unrelated to GA, of *Gmuid1-1* to H12.

Minor points:

- Line 24: "gibberellic acid (GA)"; consistent line 58.

Response: Thank you for your suggestion. We have revised as you suggested (please see the line 24).

- Line 36: What is meant by "..., effectively improving GmUID1 function."; not clear, please rephrase.

Response: Thank you for your suggestion. We exchanged "..., effectively improving GmUID1 function." to "Our results revealed that GmUID1 mediated UV-B tolerance to maintain growth in soybean by modulating GA metabolism." (Please see lines 35-36).

- lines 924-1101: Several references listed twice; please recheck.

Response: Thank you for your suggestion. We have removed multiple duplicates of references (please see the lines 1078-1251).

- Line 77: " UV RESISTANCE LOCUS8 (UVR8)"

Response: Thank you for pointing out this problem. We have exchanged "UVB RESISTANCE8 (UVR8)" to "UV RESISTANCE LOCUS8 (UVR8)" (please see the line 76).

- Line 110: "The other E3 ubiquitin ligase, ..." suggests there are only two E3 ligases in soybean. Please rephrase.

Response: Thank you for your suggestion. We exchanged “The other E3 ubiquitin ligase, ...” to “Another E3 ubiquitin ligase, ...” (please see the line 109).

- Line 114: "affection"? Entire sentence not clear to me...

Response: Thank you for your suggestion. We rephrased the sentence “Our study characterized the effect of UV-B exposure and GA application on plant height” (please see the line 113).

- Line 119/120: "... might be due to the damage of cis-regulatory elements". Rather: due to specific mutations in cis-regulatory elements?

Response: Thank you for your suggestion. We exchanged “... might be due to the damage of cis-regulatory elements” to “due to the specific mutations in cis-regulatory elements” (please see the lines 118-119).

- Line 141: "..., 153 showed the wild-type phenotype"; add "wild type"?

Response: Thank you for your suggestion. We added “wild type” (please see the line 139).

- Line 165: "new increased exon"? "exon of the protein"? Please correct.

Response: Thank you for your suggestion. We rephrased the sentence “increased exon of the transcript” (please see the line 163).

- Fig. 4c: why the smear also below the band if indicating ubiquitination?

Response: Thank you for your suggestion. We re-performed the *in vitro* ubiquitination assay for GmGA2ox-like-MBP. The new results clearly showed that GmGA2ox-like can be ubiquitinated (please see the new Fig. 4c).

- Line 315: should likely read "... under white light in the field (Fig. 5c-d)". In general, please describe conditions clearer in figure legends.

Response: Thank you for your suggestion. We rephrased the sentence “The results showed that exogenous GA3 treatment restored the height of *Gmuid1-1* to that of H12 when grown under sunlight” and added the conditions clearer in figure legends (please see lines 336-337, the new

Fig.5a).

- Line 321: in contrast to statement "... within 12 h of combined treatment with GA3 and UV-B light (Fig. 5g-h)" apparently no combined treatment shown in Fig. 5g-h.

Response: We do appreciate your carefully checking and thank you for pointing out this wrong description. We rephrased the sentence "within 12 h treatment with GA3 or UV-B light" (please see the line 361).

- Fig. 5j: n=? Field experiment?

Response: Thank you for your suggestion. We added the description in figure legends. "Number of $GmUID1^{Hap1} = 201$, number of $GmUID1^{Hap2} = 38$, number of $GmUID1^{Hap5} = 81$, number of $GmUID1^{Hap6} = 84$ ". And The data was obtained through field statistics.

- Lines 454-454: "... relative GmUID1 transcript activity were higher in accessions harboring...?"

This statement seems to refer to GUS assays in *N.benthamiana*... Needs to be corrected.

Response: Thank you for your suggestion. We rephrased the sentence "relative GmUID1 transcript activity was higher harboring Hap1, Hap2 and Hap6 compared to those with Hap5 upon exposure to UV-B in the *N.benthamiana* transient expression system" (please see the lines 513-514).

- Line 472: Light and growth conditions? How old plants? How long exposed to UV-B?

Response: Thank you for your suggestion. It has been revised in figure legends of new Extended Data Fig. 12.

- Fig. 2 b-h. Conditions are not clearly described, but it seems 2b very different to 2d-h. Re-arrange figure that one does not compare plant height in 2b versus others measurements. As they are directly next to each other, it is rather confusing. Mention the light treatments in the legend (duration, fluence rate).

Response: Thank you for your suggestion. It has been revised in the new Fig. 2c.

- Fig. 4a: why is there such a strong difference in the input between – and + GmGA2ox-like-GFP

on total Ub-Flag?

Response: Thank you for your suggestion. We had repeated the experiment, there was no difference in the input between – and + GmGA2ox-like-GFP on total Ub-Flag, please see the new Fig. 4a.

- Lines 739 – 740: provide fluence rates in $\mu\text{mol m}^{-2} \text{sec}^{-1}$.

Response: It has been revised as you suggested.

Reviewer #3 (Remarks to the Author):

Sun et. al. provides a new hypothesis on the post-translational regulation of GA2ox-like in a UVB-dependent manner in Soybean. This is an intriguing possibility which was not been explored before. The authors also tried to provide evidence for the E3 ligases that ubiquitinate the GA2ox-like. In addition, the prevalence of different haplotypes in one of the E3s, the GmUID1 has been investigated and proposed that a defect in its cis-element may have contributed to a UVB-dependent plant height phenotype.

Even though the manuscript provides a large amount of experimental data, I have some serious concerns over the ways in which the experiments were conducted and the rationale behind the explanation and interpretation of the results. I have the following major and minor comments that I request the authors to address in detail:

Major:

1. The authors provide *Gmilpa1* as a mutant allelic to *Gmuid1-1*. *Gmilpa1* has been published previously (ref. 53 in the manuscript). However, *Gmilpa1* do not show a dwarf phenotype (Figure 1 and 3 of ref. 53). Can the authors please clarify this discrepancy?

Response: Thank you for your comments. As we noticed, the plant height of *Gmilpa1* was shorter than that of H12 as reported in the Supplemental Figure 3 of the previous publication (Gao et al Plant Physiol. 2017). In addition, we found that *Gmilpa1* was also reported in an earlier publication (Cheng et al Journal of Integrative Agriculture. 2016), and found that all mutants reported in this manuscript are all characterized by dwarfism (Cheng et al Journal of Integrative Agriculture. 2016). Furthermore, when we planted *Gmilpa1* in the field, it showed a dwarfing phenotype (please see

Fig. 1e), and F₁ plants of *Gmuid1-1* and *Gmilpa1*(*Gmuid1-2*) crossing lines also exhibited the dwarf phenotype in the field (please see Fig. 1e). So, *Gmilpa1* (*Gmuid1-2*) is a mutant allele of *Gmuid1-1*.

2. Even though the authors show data for the mutual interaction between GmUID1, GmUBL1 and GmGA2-ox-like, it is not clear how these interactions ultimately cooperate in GmGA2ox-like degradation in presence of UVB. There are several questions remain unanswered:

a. What is the role of GmUBL1 in GmUID1-GmGA2ox-like interaction influenced by UVB?

Response: Thank you for your comments. Our results showed that GmUID1 mediated the degradation of GmGA2ox-like; GmUBL1 could not mediate the degradation of GmGA2ox-like; however, the degradation of GmGA2ox-like is accelerated in presence of GmUBL1 (Fig. 4). Therefore, we suggested that GmUBL1 enhanced the GmUID1-mediated degradation of GmGA2ox-like.

b. What are the respective contributions of GmUBL1 and GmUID1 in ubiquitinating GmGA2ox-like?

Response: We found that GmUBL1 and GmUID1 both ubiquitinated GmGA2ox-like, respectively. However, GmUBL1 alone could not induce the degradation of GmGA2ox-like, but promoted GmUID1-mediated degradation of GmGA2ox-like. Therefore, we predict that GmUID1 mediated the stabilization of GmGA2ox-like, which were fine-tuned by GmUBL1.

c. What is the relevance of direct interaction between GmUBL1 and GmUID1?

Response: As GmUBL1 promoted GmUID1-mediated degradation of GmGA2ox-like, we suggest that the interaction between GmUBL1 and GmUID1 might form a larger E3 ubiquitin ligase complex to regulate the protein stability of GmGA2ox-like. In plants, several studies have showed that different E3 ligases could interact with each other to form an E3 complex to mediate the ubiquitination of their substrates (Huang, X et al PNAS 2013; Ren, H et al PNAS 2019). Therefore, we predict that GmUBL1 and GmUID1 might form or belong to a E3 ligase complex to regulate the ubiquitination and stability of GmGA2ox-like, which deserves further investigation.

3. The experiments involving different wavelengths of light are not convincing.

a. UVB is often detrimental to plant growth, so weak narrow-band UVB coupled with white light is preferred, especially at the seedling stage. However, the authors have used intense UVB radiation alone (methods section; Lines 739-741). This could induce potential stress on the plants. Moreover, the exact experimental conditions used are unclear.

Response: We do agree that weak narrow-band UVB coupled with white light is better for our research. Based on your comments, we performed the phenotypic analysis experiments of H12 and *Gmuid1-1* under white light and weak narrow-band UV-B coupled white light. And we found that the results were consistent with those of our previous experiments (please see the new Fig. 2c).

b. For Fig. 2C-2H: The quantity, duration and frequency of light treatment were not mentioned. It is also important to mention the ZT at which the light treatments were given, as this affects the photoreceptor activities.

Response: Thank you for your suggestion. (1) We added the detailed description of quantity, duration and frequency treated with UV-B condition in the figure legends and methods, the description was as following: H12, *Gmuid1-1*, and *Gmuid1-2* grown under white light (600 $\mu\text{mol m}^{-2} \text{s}^{-1}$ white light, no UV-B, 25°C, and a 12-h-light/12-h-dark photoperiod) and white light supplemented with UV-B (600 $\mu\text{mol m}^{-2} \text{s}^{-1}$ white light, 1.5 $\mu\text{mol m}^{-2} \text{s}^{-1}$ UV-B, 25°C, and a 12-h-light/12-h-dark photoperiod) (please see the new Fig. 2c lines 544-547). (2) This revised photoperiod condition does not affect photoreceptor activity.

c. The methods section (Lines 739-742) shows that the UVB treatment was only for 2 hours and the plants were returned to white light. Irradiation with the other wavelength is also apparently performed in a similar fashion. It is surprising that such a short treatment was able to produce the seedling phenotypes shown in Fig 2C. Again, the experimental conditions are not well explained.

Response: Thank you for your suggestion. According to your suggestion, we changed the UV-B treatment condition to weak narrow-band UV-B coupled white light, which was commonly used in the UV-B signaling research field (Podolec, R. et al PNAS. 2020; Podolec, R. et al The plant journal. 2022) and added the phenotypes of H12 and *Gmuid1-1* under white light and white light coupled with weak narrow-band UV-B. The results are consistent with those in the previous version (please

see new Fig. 2c).

d. It is not clear how the UVB experiments were performed for Figures 4i, 5e, 5h, 6h, Ext. data Fig. 7 and Ext. data Fig. 8

Response: Thank you for your suggestion. The methods of UV-B experiments for Figures 4i, 5e, 5h, 6h, Ext. data Fig. 7 and Ext. data Fig. 8 were described in detail in the figure legends and the section of “Methods” (please see the new Fig. 4i, Fig. 5c, Fig. 5h, Fig. 6h, Ext. data Fig. 7 and Ext. data Fig. 10).

4. There are multiple problems with the protein interaction experiments

a. Lane 4 and Lane 8 of the Y2H data in Fig. 3A: Seem to be duplicated

Response: Thank you for your comments. We reformed Y2H experiment and exchanged it to the new Fig. 3a. The results showed that GmUID1 interact with GmGA2ox-like and GmUBL1 respectively, and GmUBL1 also interact with GmGA2ox-like.

b. The authors have used 3-AT presumably because the BD-GmUID1 showed autoactivation. A fold-dilution series should be performed in this scenario to show the interaction strength. Without this, the results can only be interpreted as a consequence of autoactivation

Response: Thank you for your suggestion. According to your suggestion, the fold-dilution series was performed in Y2H in revised manuscript, please see the new Fig. 3a.

c. Using the yeast system, the UVB-dependency of the interactions could easily be tested. I suggest the authors perform the Y2H interaction assays in a quantitative manner in the presence or absence of UVB

Response: Thank you for your suggestion. We used Co-IP instead of Y2H experiment in the presence or absence of UVB, as also suggested by Reviewer 2. The results showed that UV-B enhanced the interaction between GmUID1 and GmGA2ox-like (please see the new Fig. 3h). However, UV-B treatment failed to affect the interaction between GmUID1 and GmUBL1 or between GmGA2ox-like and GmUBL1 (please see the new Fig. 3h, Ext. data Fig. 7f, 7i).

d. Co-IP experiments were performed in *Nicotiana*, while the authors already have transgenic lines expressing GFP-tagged proteins. These lines could be used effectively coupled with agrobacterium-infiltration (King et. al., 2015) for all the Co-IP and protein degradation experiments shown in the manuscript

Response: Thank you for your suggestion. We performed the Co-IP experiments with anti-GmUID1 in *35S:GmGA2ox-like-GFP*, the result showed that GmUID1 interact with GmGA2ox-like in soybean (please see the new Ext. data Fig. 7c).

e. Moreover, the authors have produced specific antibodies that detect the Gm proteins. Then why use tagged proteins in *Nicotiana*?

Response: Thank you for your suggestion. We performed the Co-IP experiments with anti-GmUID1 in *35S:GmGA2ox-like-GFP* (please see the new Ext. data Fig. 7c).

f. Equal loading controls for the input samples are missing in the Co-IP experiments

Response: Thank you for your suggestion. we repeated the Co-IP experiments in *Nicotiana* and added the equal loading controls for the input samples (please see the new Fig. 3c-e).

g. Co-IP experiments were shown in chopped images. Hence the original blots must be provided as additional data

Response: Thank you for your suggestion. We provided the original blots in the source data.

h. In general, proper controls are missing from BiFC experiments. Ideal experiments must include positive and negative controls that improve reliability (Horstman et. al., 2014)

Response: Thank you for your suggestion. For BiFC experiments, mutated version of GmGA2ox-like serves as a negative control for the interaction between GmUID1 and GmGA2ox-like, GmPUB21(U-Box E3 Ubiquitin Ligases, reported by Yunhua Yang et al 2022) serves as a negative control for the interaction between GmUID1 and GmUBL1, GmUBL1 and GmGA2ox-like respectively. The negative control had no fluorescence (please see the new Fig. 3b).

i. The strength of interaction of GmGA2ox-like and GmUID1 under white light in Fig. 3C is much

stronger than the one shown in Fig. 3H. Can the authors comment on this?

Response: Thank you for your suggestion. As the brightness in the Fig. 3c were adjusted, the interaction intensity between GmGA2ox-like and GmUID1 in Fig. 3c seemed much stronger under white light than that in Fig. 3h. In the revised manuscript, we used the same settings to adjust the brightness for the images in Fig. 3c and Fig.3h.

j. In Fig. 3H, the method of quantification, laser intensity and exposure time used, UVB treatment and duration should be provided

Response: Thank you for your suggestion. The method of quantification, laser intensity, exposure time used, UVB treatment and duration was added in the figure legends and “methods” (“The relative fluorescence intensities of cytoplasm and whole cells were quantified and the cytoplasm-to- background ratios are plotted, *N. benthamiana* leaves were co-infiltrated with GmGA2ox-like-nYFP and GmUID1-cYFP, and exposed to 1 h of UV-B ($21\mu\text{mol m}^{-2} \text{s}^{-1}$) before imaging”), please see the new Fig. 3f-g and lines 928-930.

k. Figure 3H-3I: It is not clear why the interaction has been observed only in the cytoplasm and not in the nucleus, since GmUID1 also localizes strongly in the nucleus (Ext. Fig. 5C). Also, the subcellular localization of GmUBL1 and GmGA2ox-like was also not shown. Subcellular localization of all three proteins would help in interpreting the data shown in Fig. 3i

Response: Thank you for your suggestion. Co-localization of GmGA2ox-like and GmUID1, GmUBL1 and GmUID1 were performed, the results indicated that GmUID1 and GmGA2ox-like, GmUID1 and GmUBL1 were co-localized in cytoplasm and nucleus by transient expression in *N. benthamiana*, please see the new Extended Data Fig. 8.

5. Discrepancies in the ubiquitination assay. The experiments are not convincing enough for establishing GmGA2ox-like as a substrate of GmUID1

a. The authors resort to using *Nicotiana*, while they have created Gm transgenic plants. Please see also comment 5a

Response: Thank you for your suggestion. The GFP-tagged GmGA2ox-like lines were used for ubiquitination assay, the result showed that GmGA2ox-like can be ubiquitinated in soybean plants

(please see the new Fig. 4f).

b. Fig. 4a: The misexpressed Gm proteins are shown to be ubiquitinated in Nicotiana. Are there any orthologues in Nicotiana that act as E3? If so, is it also UVB-dependent? Ideally, GmUID1 should also be co-infiltrated to test ubiquitination. Alternatively, the GFP-tagged Gm line should be used for ubiquitination assay

Response: Thank you for your suggestion. (1) We performed ubiquitination assay by using co-infiltrated GmUID1 and GmGA2ox-like to test ubiquitination of GmGA2ox-like, the result showed that there was a remarkably increased smear representing poly-ubiquitinated GmGA2ox-like in the presence of GmUID1, please see the new Fig. 4d. (2) We also performed ubiquitination assay by using the GFP-tagged GmGA2ox-like lines, the result showed that GmGA2ox-like can be ubiquitinated in soybean plants (please see the new Fig. 4f).

c. Authors have used an anti-Ub antibody in Fig. 4B but infiltrated Ub-FLAG in Figure 4A. Why not use anti-Ub in both experiments?

Response: Thank you for your suggestion. We used anti-flag antibody in Fig.4a because Ub-Flag is a recombinant purified protein, and anti-flag antibody can also detect the ubiquitination. For Fig.4b, we used anti-Ub antibody to detect the ubiquitination level of GmGA2ox-like in Nicotiana.

d. Fig. 4A: Blot with anti-GFP should also be shown. Loading controls are missing for the input samples. As ubiquitinated proteins are degraded while doing the experiment, Loading controls experiments, MG132 is often used. It is not clear how this experiment was performed in Nicotiana

Response: Thank you for your suggestion. (1) We re-performed western blot using anti-GFP and added the results to the new Fig. 4a. (2) MG132 was added in the experiment conducted by Nicotiana, and we added the description in the method of the revised manuscript (please see line 948, line 980).

e. Fig. 4B: The input samples in lanes 3 and 4 seem to be low and that may be the reason for the lower ubiquitination.

Response: Thank you for your comments. We re-performed western blot for Fig. 4b, and the results

showed that the input sample abundance of each lane was similar (please see the new Fig. 4b).

f. Fig. 4C: GmUID1 seems to be auto-ubiquitinated. However, this is not addressed anywhere in the manuscript. To have more reliability, the K394R version should be included in the *in vitro* ubiquitination assay

Response: Thank you for your suggestion. (1) The related description of auto-ubiquitinated of GmUID1 was added in revised manuscript (Lines 270-271). (2) The K394R version was included in the *in vitro* ubiquitination assay. The result showed that GmGA2ox-like^{K394R} cannot be ubiquitinated, please see the new Fig. 4c.

g. Fig. 4D: There is no explanation of the ways in which the experiment was conducted. It is not clear why the time points were shown or what to expect from this figure

Response: Thank you for your suggestion. Due to the GmGA2ox-like degradation experiments have been conducted in soybeans (please see the new Fig. 4i) and was repeated in *Nicotiana* (please see the new Fig. 4j), therefore, we deleted the Fig. 4d in revised manuscript.

h. It is surprising that the GmGA2ox-like-GFP infiltrated in *Nicotiana* shows strong ubiquitination in Figure 4A, while it shows weak ubiquitination in Figure 4e. How would the authors explain this?

Response: Thank you for your comments. In Fig. 4a, we used anti-flag to detect the ubiquitination level of GmGA2ox-like, In Fig. 4e, we used anti-Ub antibody to detect the ubiquitination level of GmGA2ox-like. Therefore, we thought that the difference between Fig. 4a and Fig. 4e was due to differences in antibodies.

i. Fig. 4F and 4G: The details of UVB experiments are missing

Response: Thank you for your suggestion. We added the details of UVB experiments in the figure legends and methods “Immunoblots showing GmGA2ox-like protein levels in V2 stage seedlings of H12 and *Gmuid1-1* grown in white light ($600 \mu\text{mol m}^{-2} \text{s}^{-1}$, 12 h/12 h light/dark) and then transferred to white light supplemented with UV-B ($1.5 \mu\text{mol m}^{-2} \text{s}^{-1}$, 12 h/12 h light/dark) for the indicated time periods” (please see the new Fig. 4i).

j. Fig. 4i and 4J: Complete blots should be provided as additional data

Response: Thank you for your suggestion. We provided all original blots in the source data.

k. In Fig. 4J, blot at the end of the right-hand side, last lane: This is important data that shows the degradation of GmGA2-ox-like. However, the Ponceau staining shows that fewer proteins were loaded in this well

Response: Thank you for your suggestion. According to your comments, Fig. 4J had been reperformed with the same amount of proteins loaded (please see the new Fig. 4j).

l. Fig. 4i and 4J: Since the protein degradation is apparent from 90 minutes, the time points for the experiment should be extended up to 150 minutes, which will show the degradation dynamics. This is especially relevant since the UVB-mediated enhancement in GmUID1 could be observed only after at least 3-4 hours of UVB exposure (Fig. 5G)

Response: Thank you for your suggestion. Indeed, UVB-mediated enhancement in GmUID1 could be observed after at least 3-4 hours of UVB exposure (Fig. 5g, new Fig. 5l), therefore, we tested the protein level of GmGA2ox-like in H12 and *Gmuid1-1* with and without UVB, the time points were extended up to 12 hours, the result showed that GmGA2ox-like showed substantial degradation in WT exposed to UV-B from 10 h, but not in *Gmuid1-1* under the same conditions, see the new Fig. 4i. Fig.4i of previous manuscript was moved to the new Extended Data Fig. 9. For Fig.4j, the time points were extended up to 150 minutes, the result showed that the protein degradation is apparent from 60 minutes to 150 minutes (please see the new Fig. 4j).

m. Since GmUID1 targets the D-BOX in GmGA2ox-like, a D-BOX mutated version should be used for the ubiquitination assays in order to verify that GmUID1 acts as an E3 in degrading GmGA2ox-like

Response: Thank you for your suggestion. The D-BOX mutated version was used for the ubiquitination assays in Nicotiana, the results showed that GmUID1 cannot enhance the ubiquitination of D-BOX mutated version (see the new Extended Data Fig. 9a).

6. For the UVB-induced phenotype in Fig. 2C-2H, only *Gmuid1-1* is used. Is there any specific

reason? Since the UVB phenotype is the basis for further experiments in the manuscript, the UVB phenotype of *Gmuid1-2/Gmilpa1* and also the F₁ plants resulting from the cross *Gmuid1-1* x *Gmuid1-2* should be used. This is required to establish the role of *GmUID1* as an enhancer of plant height under UVB.

Response: Thank you for your suggestion. (1) The UV-B-induced phenotype of *Gmuid1-2/Gmilpa1* was added in revised manuscript. The results showed that the plant height of *Gmuid1-2/Gmilpa1* under UV-B was similar with that of *Gmuid1-1* (please see the new Fig. 2-d). (2) F₁ plants the *Gmuid1-1 Gmilpa1* crossing lines also exhibited the dwarf phenotype in the field (Fig. 1e).

7. To complement the *Gmuid1-1* mutant, the authors resorted to overexpressing the full-length cDNA tagged with GFP. Ideally, the native promoter driving full-length genomic DNA should be used for complementation.

Response: Thank you for your suggestion. Indeed, the native promoter driving full-length genomic DNA is better for complementation, we have not yet obtained the transgenic line. In our study, the plant height of *Gmuid1-2/Gmilpa1* under UV-B was similar with that of *Gmuid1-1* (please see the new Fig. 2-d), and F₁ plants from the cross between *Gmuid1-1* and *Gmilpa1* also exhibited the dwarf phenotype. These results demonstrated that mutation of *GmUID1* caused the dwarf phenotype in *Gmuid1-1*.

8. Specific antibodies used in experiments such as the ones shown in Fig. 4 need validation experiments. The authors could easily show this via western blots using the WT, overexpression lines and mutants. What protein sizes are detected by these antibodies? Are there any non-specific bands detected? Please provide this as additional data

Response: Thank you for your suggestion. We performed western blots using the WT, overexpression lines and mutants to verify the specific antibodies. The results indicated that the antibody had specific bands (please see the new Extended Data Fig. 7a-b).

9. In Fig. 5A and 5B: GA51 content was not registered in the WT. Normally a basal level of GA degradation is expected in WT. Can this be explained?

Response: Thank you for your suggestion. We again measured the contents of endogenous GAs

(GA1, GA4, GA20, GA9, GA8 and GA34) in H12 and *Gmuid1-1* under white light and white light supplemented with UV-B conditions, the results showed no difference in GAs content between H12 and *Gmuid1-1* under white light (Fig. 5e-j), while the gibberellin GA1, GA4, GA20 and GA9 were more abundant in the WT than in *Gmuid1-1* under white light supplemented with UV-B, the non-bioactive GA8 and GA34 accumulated at a relatively high level in *Gmuid1-1* relative to the H12 (please see the new Fig. 5e-j). However, GA51 was not detected in H12 and *Gmuid1-1*, probably due to the extremely low content.

10. Fig. 5E: Please clarify why PAC along with GA was used in this UVB experiment

Response: Thank you for your suggestion. GA biosynthesis inhibitor paclobutrazol (PAC) efficiently inhibited the elongation of H12, *Gmuid1-1* and *Gmuid1-2* (Fig. 5c-d), and the exogenous application of GA₃ rescued the PAC-induced growth deficiency of H12, *Gmuid1-1* and *Gmuid1-2* seedlings excluded side effects of PAC, therefore, PAC along with GA was used in this UVB experiment which confirm that GA can restore the dwarf phenotype of the mutant under UVB conditions (please see new Fig. 5c).

11. Fig. 6H: The difference in GUS activity seems to be rather due to different infiltration efficiency.

There are no infiltration controls provided in this experiment

Response: Thank you for your suggestion. To eliminate the infiltration efficiency, we performed qPCR on the gene *bar* on the GUS vector as infiltration controls (please see the new Extended Data Fig. 12a). The results showed that the differential transcriptional activities between *Hap5* and other haplotypes under UV-B expose is caused by the deletion of CRE (*cis*-regulatory elements) in the *GmUID1* promoter of *Hap5*, which has a significant effect on the response to UV-B.

12. For the GUS assay of haplotypes, an ideal control could be hap5 with the INDEL. Moreover, it would be interesting to see the phenotype of hap5 in the presence or absence of UVB

Response: Thank you for your suggestion. (1) We generated the construct of *GmUID1*^{Hap5} promoter with indel *GmUID1*pro^{Hap5/indel}:GUS, and transiently expressed each construct in *N. benthamiana* leaves via *Agrobacterium*-mediated infiltration to test their transcriptional output with and without UV-B treatment. Histochemical staining showed no difference in GUS signal among

GmUID1pro^{Hap1}:GUS, *GmUID1pro^{Hap3}:GUS*, *GmUID1pro^{Hap5}:GUS*, *GmUID1pro^{Hap5/indel}:GUS* and *GmUID1pro^{Hap6}:GUS* when infiltrated leaves were maintained under white light (Fig. 6h). We did not observe a difference in GUS signal for *GmUID1pro^{Hap5}:GUS* regardless of UV-B exposure (Fig. 6h). However, the GUS signal for *GmUID1pro^{Hap1}:GUS*, *GmUID1pro^{Hap3}:GUS*, *GmUID1pro^{Hap5/indel}:GUS* and *GmUID1pro^{Hap6}:GUS* appeared higher than that of *GmUID1pro^{Hap5}:GUS* when the infiltrated leaves were exposed to UV-B (Fig. 6h). We confirmed these results with a quantitative GUS assay (Fig. 6i). (2) H12 belongs to hap5, the phenotype of H12 in the presence or absence of UVB was showed in the new Fig. 2c.

13. Figure 6J: The plant height phenotype of haplotypes should be provided

Response: Thank you for your suggestion. In the revised version we had provided the data for plant height of different haplotypes.

14. Ext. data fig. 1C and 1D: The number of replicates and the number of cells per replicate used for the experiment should be stated. Microscopy procedures should be added to the methods section

Response: Thank you for your suggestion. The number of replicates and the number of cells per replicate used has been added in the figure legends (The number of cells was from three independent replicates of 50 cells.), please see the new Extended Data Fig. 1d. Microscopy procedures have been added to the methods (please see the lines 932-940).

Minor:

1. Lines 26-27: APC/C is abbreviated in the abstract without providing the full form

Response: Thank you for your suggestion. It has been revised as you suggested. We exchanged “APC/C” to “Anaphase Promoting Complex/Cyclosome (APC/C)” (please see the line 27).

2. The effect of GA on UVB-induced hypocotyl elongation has recently been shown in Arabidopsis (Miao et. al., 2021). The results should be discussed in this context

Response: Thank you for your suggestion. The results from Miao et. al., 2021 had been added to the discussion section (please see the lines 465-466).

3. Line 114: Do the authors mean the effect of UVB and GA?

Response: Thank you for your comment. We rephrased the sentence “Our study characterized the effect of UV-B exposure and GA application on plant height” (please see the line 113).

4. Mostly the form “UV-B” is used in the manuscript. However, in some places “UVB” is used. I suggest making the word uniform throughout

Response: Thank you for your suggestion. We had replaced UVB with UV-B.

5. Include microscopy procedures followed in the methods section

Response: Thank you for your suggestion. Microscopy procedures had added to the methods (please see the lines 932-940).

6. Lines 140-141: Do the authors mean 150 plants with WT phenotype?

Response: Thank you for your suggestion. We rephrased the sentence “153 showed the wild-type phenotype and 48 exhibited the mutant phenotype” (please see the line 139).

7. Lines 154-155: The sentence lack information on NGS used in the bulk segregant analysis

Response: Thank you for your comment. Information on NGS had been added in revised manuscript as you suggested (please see the line 156).

8. In Fig. 3, C, E and G could be combined with common controls

Response: Thank you for your suggestion. Fig. 3, C, E and G had combined with common controls (please see the new Fig. 3b).

9. Line 252: The bioinformatic tool used should be specified in the methods section

Response: Thank you for your suggestion. The bioinformatic tool used had been added in methods (please see the lines 971-973).

10. Line 295: Space between proteasome and in

Response: Thank you for your suggestion. It has been revised as you suggested.

REVIEWER COMMENTS

Reviewer #1 (Remarks to the Author):

The revised manuscript includes substantially more data to support the conclusions. I have still some reservations, but overall the work is of considerable interest, demonstrating post-translational regulation of a GA-biosynthetic enzyme. However, the mechanism for promotion of the association of GA2ox-like with GmUID1 is unclear. As noted previously the effect on stature of activating this enzyme in the gmuid1 mutant is much more striking than would be anticipated and although height was restored by treating with gibberellin, the plants look very different from WT plants. This could be due to the dose of GA, and a more concerning rescue, albeit partial, was obtained by RNAi knock-down of GA2ox-like.

The identity of the GA2ox-like gene is still uncertain. The phylogenetic analysis in supplementary Fig. 6 is strange and does not agree with many other published analyses of these proteins. GA2ox4 is a close paralogue of GA2ox6 (AT1G02400), while GA2ox7 (AT1G50960) and GA2ox8 (AT4G21200) belong in a separate clade from the others, including GA2ox4. The authors should check this carefully. Ideally, the function of GA2ox-like should be confirmed biochemically.

As GmILPA1 was published first, it would seem more reasonable to use this name for the new mutant allele.

More information is provided for the method used for gibberellin analysis, but it is not complete. Importantly the identity and amounts of internal standards should be provided as well as the ions monitored.

Minor points:

Lines 143 and 150: should be markers.

Line 186: "both the nucleus and cytoplasm" repeated.

Line 327: replace "while" with "but"

Lines 359-361: As GmUID1 transcript levels do not respond to UV-B, why is the observation that GmUID1 accumulated under UV-B consistent with this?

Reviewer #2 (Remarks to the Author):

This is a revised version of a manuscript that I reviewed before. The authors satisfactorily addressed my previous concerns and have improved the manuscript significantly.

I only have a concern regarding new figure 4d, described on Lines 277-278 that the authors should address: This experiment misses a control showing that what is detected is indeed ubiquitinated GmGA2ox-like-GFP and not auto-ubiquitinated GmUID1! Line 271 and 4c mentions this feature of GmUID auto-ubiquitination. This is crucial as GmUID1-MYC is co-IPed with GmGA2ox-like-GFP, and from the provided data it can not be distinguished whether the smear in the anti-ubiquitin western is due to GmGA2ox-like-GFP or GmUID1-MYC ubiquitination. A similar control is missing in Fig.4e (although in contrast to GmUID1 auto-ubiquitination of GmUBL1 is not shown, it can be expected, or at least not be excluded).

Otherwise I only have a few minor suggestions and corrections:

- I think the abstract is still misleading, particularly due to the statement "..., raising bioactive GAs contents and promoting stem elongation" (reading abstract it indicates GA levels higher under UV, not that "raising contents" is relative to the UV-decreased amounts; i.e. simply less, i.e. buffered, reduction of bioactive GA). It can be thus made clearer that bioactive GAs are reduced upon UV-B contributing to growth inhibition (apparently independently of GmGA2ox-like catabolic activity), and that GmUID1 accumulation under UV-B and destabilization of GmGA2ox-like counteracts UV-B-mediated reduction of bioactive GA. The authors may consider adding a model summarizing their

finding (potentially increasing impact of the work) – but up to the authors.

Also lines 107 – 112: “We demonstrate that UV-B induces the accumulation of GmUID1, which regulated plant height by interacting with GmGA2ox-like and targeting it for degradation in response to UV-B. ... Our findings indicated that GmUID1 promotes GA catabolism to modulate plant height under UV-B exposure” – it would be helpful for the understanding if the authors would be more explicit... e.g. instead of “...regulated plant height...” maybe something like “...reduces/counteracts UV-B-induced growth inhibition by...”

- Line 172: “we named the mutant allele Gmilpa as Gmuid1-2”

- Line 197: “supplemented” instead of “coupled”

- Line 441: ref. 32 should be changed to 33, ref 60 seems to have nothing to do with the statement made. Should be rather Rizzini et al 2011 as it introduces UVR8 monomers.

Reviewer #3 (Remarks to the Author):

The authors have addressed most of my comments in a satisfactory manner. The manuscript has been significantly improved by the additional experiments, which were not trivial to perform. Further experiments could be conducted to elucidate the relative roles of GmUBL1 and GmUID1 in ubiquitinating GmGA2ox-like and the significance of their direct interaction, but these could be reserved for future work.

Reviewer#1(Remarks to the Author):

The revised manuscript includes substantially more data to support the conclusions. I have still some reservations, but overall, the work is of considerable interest, demonstrating post-translational regulation of a GA-biosynthetic enzyme. However, the mechanism for promotion of the association of GA2ox-like with GmUID1 is unclear. As noted previously the effect on stature of activating this enzyme in the gmuid1 mutant is much more striking than would be anticipated and although height was restored by treating with gibberellin, the plants look very different from WT plants. This could be due to the dose of GA, and a more concerning rescue, albeit partial, was obtained by RNAi knock-down of GA2ox-like.

The identity of the GA2ox-like gene is still uncertain. The phylogenetic analysis in supplementary Fig. 6 is strange and does not agree with many other published analyses of these proteins. GA2ox4 is a close paralogue of GA2ox6 (AT1G02400), while GA2ox7 (AT1G50960) and GA2ox8 (AT4G21200) belong in a separate clade from the others, including GA2ox4. The authors should check this carefully. Ideally, the function of GA2ox-like should be confirmed biochemically.

Response: Thank you for your suggestion. 1) Sorry for the mistake we have made for the phylogenetic analysis in the previous version, and we have re-performed it (please see the new Extended Data Fig. 6a). As expected, AtGA2oxs were divided into three clades. AtGA2ox1, AtGA2ox2 and AtGA2ox3 belong to one clade, AtGA2ox4 was a close paralogue of AtGA2ox6, GmGA2ox-like is close homolog of AtGA2ox4, which belong to another clade, AtGA2ox7 and GA2ox8 belong in a separate clade from the others. 2) To confirm whether GmGA2ox-like is a functional GA2-oxidase, we conducted the *in vitro* enzymatic activity assays (Liu, C. et al Mol. Plant 2018; Sakamoto, T. et al Plant Physiol 2001). LC-MS was used to analyze the products that obtained from the enzymatic reactions for GmGA2ox-like with GA1 and GA4 as the substrates. The results showed that GmGA2ox-like converted GA1 and GA4 to their corresponding 2 β -hydroxylated products GA8 and GA34, respectively (please see the new Extended Data Fig. 6b-c).

As GmILPA1 was published first, it would seem more reasonable to use this name for the new mutant allele.

Response: Thank you for your suggestion. We had replaced *Gmuid1-1* with *Gmilpa1-2*, and replaced GmUID1 with GmILPA1.

More information is provided for the method used for gibberellin analysis, but it is not complete. Importantly the identity and amounts of internal standards should be provided as well as the ions monitored.

Response: Thank you for your suggestion. We have provided the information (please see the supplementary data 1).

Minor points:

Lines 143 and 150: should be markers.

Response: Thank you for your suggestion. We had replaced makers with markers.

Line 186: “both the nucleus and cytoplasm” repeated.

Response: Thank you for your suggestion. We had deleted the repeated sentences.

Line 327: replace “while” with “but”

Response: Thank you for your suggestion. We had replaced while with but.

Lines 359-361: As GmUID1 transcript levels do not respond to UV-B, why is the observation that GMUID1 accumulated under UV-B consistent with this?

Response: Thank you for your suggestion. Due to the fact that H12 belongs to Hap5 and had a deletion of CRE in the promoter of H12, which cannot respond to UV-B at transcription level and only promoted the accumulation of its protein. To introduce this more precisely, we rephrased the sentence “In addition, we observed that GmILPA1 accumulated within 12 h of treatment with GA3 or UV-B light” (please see the line 368).

Reviewer #2 (Remarks to the Author):

This is a revised version of a manuscript that I reviewed before. The authors satisfactorily addressed my previous concerns and have improved the manuscript significantly.

I only have a concern regarding new figure 4d, described on Lines 277-278 that the authors should address: This experiment misses a control showing that what is detected is indeed

ubiquitinated GmGA2ox-like-GFP and not auto-ubiquitinated GmUID1! Line 271 and 4c mentions this feature of GmUID auto-ubiquitination. This is crucial as GmUID1-MYC is co-IPed with GmGA2ox-like-GFP, and from the provided data it can not be distinguished whether the smear in the anti-ubiquitin western is due to GmGA2ox-like-GFP or GmUID1-MYC ubiquitination. A similar control is missing in Fig.4e (although in contrast to GmUID1 autoubiquitination of GmUBL1 is not shown, it can be expected, or at least not be excluded).

Response: Thank you for your suggestion. According to your suggestion, GmUID1-MYC (GmILPA1-MYC) and/or GmGA2ox-like-GFP were co-infiltrated in *N. benthamiana* leaves. Total protein extracts were immunoprecipitated using anti-myc antibody-conjugated agarose beads, followed by immunoblotting with anti-ubiquitin antibody. We observed a weak signal for GmUID1-MYC (GmILPA1-MYC) auto-ubiquitination when GmUID1-MYC (GmILPA1-MYC) was expressed alone, while a remarkably increased-smear signal was detected when GmUID1-MYC (GmILPA1-MYC) and GmGA2ox-like-GFP were co-expressed, demonstrating the ubiquitination of GmGA2ox-like by GmUID1-MYC (GmILPA1-MYC) (please see the new fig. 4d). Moreover, we obtained similar results when GmUBL1-MYC and GmGA2ox-like-GFP were co-infiltrated in *N. benthamiana* leaves (please see the new Fig. 4e).

Otherwise I only have a few minor suggestions and corrections:

- I think the abstract is still misleading, particularly due to the statement “..., raising bioactive GAs contents and promoting stem elongation” (reading abstract it indicates GA levels higher under UV, not that “raising contents” is relative to the UV-decreased amounts; i.e. simply less, i.e. buffered, reduction of bioactive GA). It can be thus made clearer that bioactive GAs are reduced upon UV-B contributing to growth inhibition (apparently independently of GmGA2ox-like catabolic activity), and that GmUID1 accumulation under UV-B and destabilization of GmGA2ox-like counteracts UV-B-mediated reduction of bioactive GA. The authors may consider adding a model summarizing their finding (potentially increasing impact of the work) – but up to the authors.

Response: Thank you for your suggestion. We rephrased the sentence “which suggested that GmILPA1-GmGA2ox-like module counteract the UV-B-mediated reduction of bioactive GAs” (please see the lines 35-36). We also modified the model to show that the GmILPA1-

GmGA2ox-like module could counteract UV-B-induced growth inhibition.

Also lines 107 – 112: “We demonstrate that UV-B induces the accumulation of GmUID1, which regulated plant height by interacting with GmGA2ox-like and targeting it for degradation in response to UV-B. ... Our findings indicated that GmUID1 promotes GA catabolism to modulate plant height under UV-B exposure” – it would be helpful for the understanding if the authors would be more explicit... e.g. instead of “...regulated plant height...” maybe something like “...reduces/counteracts UV-B-induced growth inhibition by...”

Response: Thank you for your suggestion. We rephrased the sentence “... which reduced UV-B-induced growth inhibition by interacting with GmGA2ox-like and targeting it for degradation in response to UV-B” (please see the line110).

- Line 172: “we named the mutant allele Gmilpa as Gmuid1-2”

Response: Thank you for your suggestion. According to the reviewer1's suggestion, we had replaced *Gmuid1-1* with *Gmilpa1-2*, and replaced GmUID1 with GmILPA1.

- Line 197: “supplemented” instead of “coupled”

Response: Thank you for your suggestion. We had replaced supplemented with coupled.

- Line 441: ref. 32 should be changed to 33, ref 60 seems to have nothing to do with the statement made. Should be rather Rizzini et al 2011 as it introduces UVR8 monomers.

Response: Thank you for your suggestion. We had changed ref.32 to 33 replaced ref.60.

Reviewer #3 (Remarks to the Author):

The authors have addressed most of my comments in a satisfactory manner. The manuscript has been significantly improved by the additional experiments, which were not trivial to perform. Further experiments could be conducted to elucidate the relative roles of GmUBL1 and GmUID1 in ubiquitinating GmGA2ox-like and the significance of their direct interaction, but these could be reserved for future work.

Response: Thank you for your comments.